



# BAROCLINIC AND BAROTROPIC INSTABILITIES IN PLANETARY ATMOSPHERES - ENERGETICS, EQUILIBRATION AND ADJUSTMENT

Peter Read[1], Neil Lewis[1], Daniel Kennedy[1], Hélène Scolan[2,1], Fachreddin Tabataba-Vakili[3,1], Yixiong Wang[1], Susie Wright[1], and Roland Young[4,1]

[1]Department of Physics, University of Oxford, Clarendon Laboratory, Parks Road, Oxford, OX1 3PU, UK
[2]Laboratoire de Mécanique des Fluides et d'Acoustique, Université Lyon, France
[3]Jet Propulsion Laboratory, Pasadena, California, USA
[4]Department of Physics & National Space Science and Technology Center, UAE University, Al Ain, United Arab Emirates

**Correspondence:** Peter Read (peter.read@physics.ox.ac.uk)

**Abstract.** Baroclinic and barotropic instabilities, are well known as the mechanisms responsible for the production of the dominant energy-containing eddies in the atmospheres of the Earth and several other planets, as well as the Earth's oceans. Here we consider insights provided by both linear and nonlinear instability theories into the conditions under which such instabilities may occur, with reference to forced and dissipative flows obtainable in the laboratory, in simplified numerical

atmospheric circulation models and in the planets of our Solar System. The equilibration of such instabilities is also of great importance in understanding the structure and energetics of the observable circulation of atmospheres and oceans. Various ideas have been proposed concerning the ways in which baroclinic and barotropic instabilities grow to large amplitude and saturate, whilst also modifying their background flow and environment. This remains an area that continues to challenge theoreticians and observers, though some progress has been made. The notion that such instabilities may act under some conditions to

adjust the background flow towards a critical state is explored here in the context of both laboratory systems and planetary atmospheres. Evidence for such adjustment processes is found relating to baroclinic instabilities under a range of conditions where the efficiency of eddy and zonal mean heat transport may mutually compensate to maintain a nearly invariant thermal structure in the zonal mean. In other systems, barotropic instabilities may efficiently mix potential vorticity to result in a flow configuration that is found to approach a marginally unstable state with respect to Arnol'd's second stability theorem. We

discuss the implications of these findings and identify some outstanding open questions.

## 1 Introduction

One of the great achievements of the past 100 years in fluid dynamics has been the development of a theory of dynamical instability, both linear and nonlinear. In a geophysical context, this has led to a quantitative understanding of a variety of phenomena, including the processes that lead to the development of large-scale energetic eddies in rotating, stratified atmospheres

and oceans. These instabilities, known as baroclinic and barotropic instabilities, are well known as the mechanisms responsible for the production of the dominant energy-containing eddies in the atmospheres of the Earth and several other planets, as well



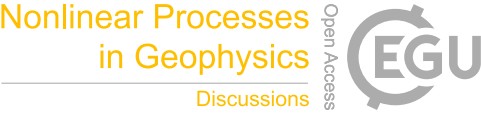

as the Earth's oceans. In general they occur in flows for which background rotation plays an important role, while baroclinic instabilities also require statically stable stratification. They are typically distinguished energetically by the respective dominance of exchanges of either kinetic or potential energy with the background (usually zonal) flow. Barotropic instability is

associated with direct exchanges of kinetic energy between background flow and eddies, while baroclinic instability entails the growth of eddy kinetic and potential energy at the primary expense of the potential energy of the background flow.

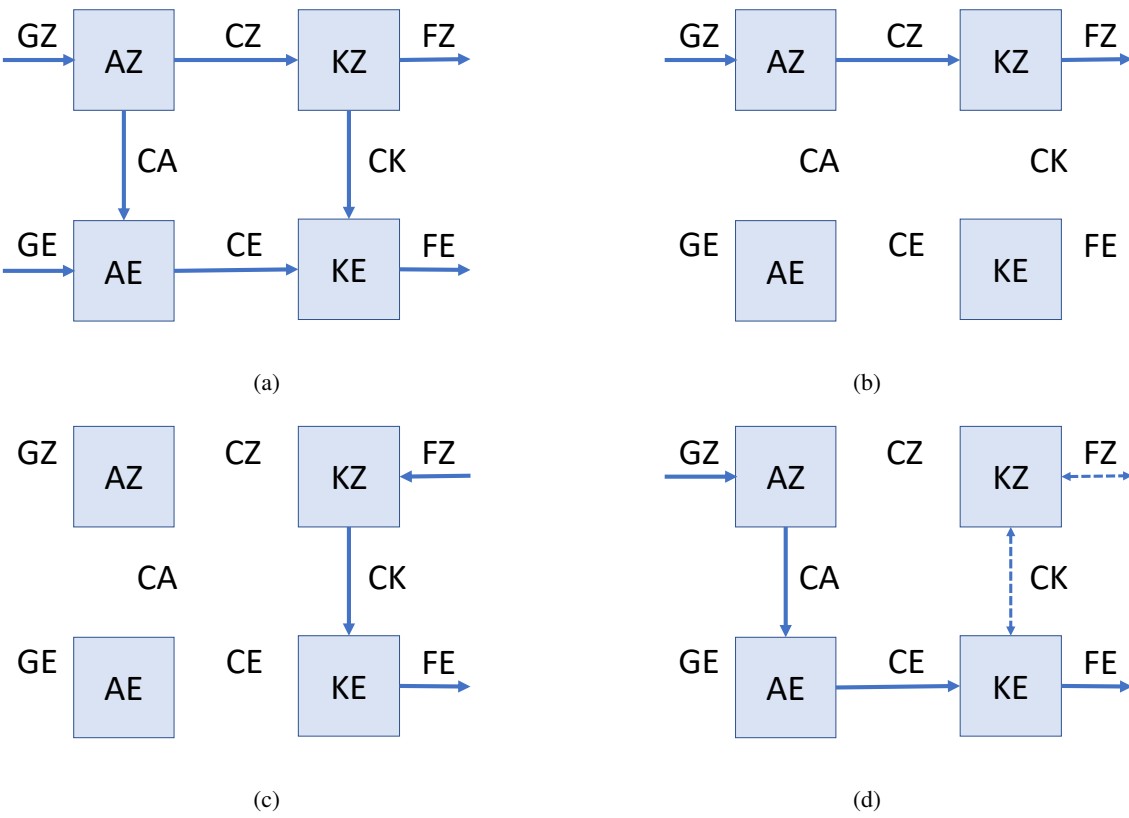

**Figure 1.** (a) Schematic representation of energy storage and interconversions in the form suggested by Lorenz (1967), with each box representing the energy reservoirs for eddy and zonal mean kinetic (KE,KZ) and available potential energies (AE,AZ) and arrows representing conversion rates between them (CZ,CK,CE,CA). Sources or sinks of potential or kinetic energy are also shown as external arrows (GZ,GE,FZ,FE). (b)-(d) are idealised examples representing the cycles expected for isolated processes (e.g. see James, 1994): (b) thermally-direct Hadley cell, (c) barotropic instabilities and (d) baroclinic instabilities.

These exchanges of energy between zonally symmetric flows and eddies, and between potential and kinetic energies, are commonly quantified in the manner originally suggested by Lorenz (1955, 1967), in which the flow is partitioned between zonally symmetric (zonally averaged) components and departures therefrom ("eddies"), with four main energy reservoirs (AE,

AZ, KE, KZ) representing respectively the eddy and zonal available potential energy, and the corresponding kinetic energies.



From considerations of the energy conservation equations, expressions can be defined to represent the rates of conversion. between the four energy reservoirs (e.g. see Lorenz, 1967; Peixoto and Oort, 1974; James, 1994), typically shown schematically in Figure 1(a).

As will be discussed further below, baroclinic or barotropic instabilities and axisymmetric overturning circulations can be distinguished energetically by differing routes of energy conversion between potential and kinetic energies. But schematically, barotropic instabilities are characterised by the dominance of the CK conversion term from zonal mean kinetic energy KZ to eddy KE (see Figure 1(c)), while energy conversions in baroclinic instabilities are dominated by the CA and CE terms, converting zonal mean potential energy AZ to eddy potential and kinetic energy, AE and KE (see 1(d)). The direct kinetic energy conversion between KE and KZ may also take place in the latter case in either direction and so is indicated by the dashed arrows in Fig. 1(d). Such circulations may also be accompanied by direct, axisymmetric overturning circulations that primarily involve the CZ conversion between AZ and KZ (e.g. see FIg. 1(b)).

In addition to straightforward energy considerations, however, the vorticity configuration (either absolute or potential vorticity) also plays a key role in determining whether an instability develops and the character of the instability when it occurs. The subsequent nonlinear development of the instability then depends also on both energetic and vorticity constraints, posing significant challenges to our understanding of how such instabilities will equilibrate.

Predicting and quantifying such equilibration processes are crucial tasks if we are to gain a full understanding of the factors that govern the observable structure and evolution of atmospheric circulation systems under a variety of different conditions. Recent exploration of our Solar System has already provided much information on the structure and properties of large- and medium-scale circulation systems across a wide range of parameter space, from very slowly rotating terrestrial planets such as Venus and Titan, through rapidly rotating terrestrial planets such as Earth and Mars to the fast rotating gas and ice giants. Baroclinic and/or barotropic instabilities likely play important roles in most of these atmospheric systems, but precisely how the occurrence and nonlinear development of such instabilities governs the structure and energetics of their circulation is still very much the subject of ongoing research, using both observations and numerical circulation models. Many planetary atmospheric circulation systems, for example, manifest complex, often anisotropic and inhomogeneous macroturbulent cascades of energy and enstrophy (squared vorticity), but how are such cascades energised and what processes determine the typical equilibrated distribution of energy? How are such states maintained by the input of thermal energy from the Sun or from the deep planetary interior?

Determining the occurrence or otherwise of either barotropic or baroclinic instabilities has traditionally appealed to the results of linearised theories, often (though not exclusively) derived from simplified forms of the equations of motion, continuity and energy conservation. The most well known approach derives from the Charney-Stern-Pedlosky (CSP) criterion for the existence of a non-zero growth rate of an infinitesimal perturbation to a background zonal flow (e.g. Vallis, 2017). This essentially reduces to the requirement that

$$-c_i \int_{y_1}^{y_2} \left\{ \frac{\partial Q/\partial y}{|U - c_r|^2} \left| \tilde{\psi} \right|^2 dz + \left[ \frac{F\partial U/\partial z}{|U - c_r|^2} \left| \tilde{\psi} \right|^2 \right]_0^H \right\} dy = 0, \tag{1}$$





where $U(y,z)$ is the background zonal flow and $Q(y,z)$ the corresponding quasi-geostrophic potential vorticity, $c = c_r + ic_i$

is the complex phase speed of the perturbation, $\tilde{\psi}(y,z)$ the amplitude of the perturbation streamfunction and $F = f_0^2/N^2$. $f_0 = 2\Omega \sin \phi_0$ is the mean Coriolis parameter, centred at latitude $\phi_0$ with $\Omega$ the planetary rotation rate, and $N$ is the Brunt-Väisälä frequency. The integrals are carried out over a domain in $(y,z)$ bounded by rigid walls or other suitable boundary conditions at $y = y_1, y_2$ and $z = 0, H$.

For $c_i \neq 0$, therefore, we require one or more of the following criteria to be satisfied.

*(i)* $\partial Q/\partial y$ changes sign in the interior domain;

*(ii)* The interior $\partial Q/\partial y$ take the opposite sign to $\partial U/\partial z$ at the upper boundary at $z = H$;

*(iii)* The interior $\partial Q/\partial y$ take the same sign as $\partial U/\partial z$ at the lower boundary at $z = 0$; or

*(iv)* $\partial Q/\partial y = 0$ in the interior and $\partial U/\partial y$ takes the same sign at both $z = 0$ and $z = H$.

Thus, the Earth's atmosphere typically satisfies criterion *(iii)*, being dominated by the planetary vorticity gradient $\beta = $

$df/dy$ away from the lower boundary and $\partial U/\partial z > 0$ at the ground, consistent with geostrophic balance and an equatorward temperature gradient. In the oceans, in contrast, either criteria *(i)* or *(ii)* may be satisfied to result in mesoscale eddies generated by baroclinic instability. But whether or how these instability criteria may be satisfied on other planets, where conditions may be very different from Earth, is often unclear, leaving open questions as to the respective roles of baroclinic or barotropic instability in the global circulation.

Even where it can be established that one or other CSP criterion can be satisfied, the fully-developed nonlinear form of the instability may also be hard to predict without the use of complex and expensive numerical models. One possible hypothesis, however, that has proved insightful and interesting in some circumstances is the suggestion that the instability grows vigorously and feeds back onto the background flow, tending to restore it to a state of marginal instability or some other well defined state. For baroclinic instabilities, this process is often referred to as *baroclinic adjustment* (e.g. Stone, 1978; Zurita-Gotor

and Lindzen, 2007; Schneider, 2007), by analogy with the concept of rapid convective adjustment towards a state of neutral static stability in a convectively unstable flow. Such a process might be expected to occur where the background unstable flow is only weakly forced or maintained, on timescales that are long compared with the advective timescale of the developing baroclinic instability. The applicability of this concept to the Earth's atmosphere was first suggested by Stone (1978), based on an application of the Phillips two-layer model of baroclinic instability. However, the validity of this application is controversial

because the Earth's atmosphere is always baroclinically unstable, even though the dominant perturbations may become much shallower than the height of the troposphere. Thus, a state of marginal instability is not well defined, leading to alternative hypotheses for the adjusted state.

In this paper we consider the application of the CSP and other stability criteria to rotating, stratified flows that include laboratory experiments and both idealised and observed planetary atmospheres. We also explore some of the properties of

the fully developed circulations with a view to quantifying the energetics of the equilibrated state and identifying the action and consequences of either baroclinic or barotropic adjustment. Section 2 reviews the results of laboratory experiments on



fully developed baroclinic and barotropic instabilities across a broad range of parameter space, examining the usefulness or otherwise of the CSP criteria and any evidence for signatures of adjustment or other forms of self-organized criticality. Section 3 reviews the corresponding properties of idealised general circulation model (GCM) simulations of an Earth-like planetary atmosphere circulation over a broad range of rotation rates, for comparison with the known properties of the atmospheric circulations of the main planetary bodies of the Solar System. We draw some conclusions and an outlook for future research in Section 5.

## 2 Baroclinic and barotropic instabilities in the laboratory

Laboratory experiments on rotating, stratified flows provide a powerful vehicle for exploring and understanding the fully developed forms of processes such as baroclinic and barotropic instability. The imposed boundary conditions and experimental parameters can often be closely controlled so that experiments can be repeated and checked, while the geometry can be kept relatively simple so that the respective influence of different factors can be tested over a wide range of conditions. Measurement techniques have also advanced considerably in recent years, allowing optical and other non-intrusive flow measurement as well as the use of in situ probes. Both baroclinic and barotropic flow configurations have been explored in past work over the past 50 or more years, with insightful results on bifurcations between steady, periodic and chaotic flow states and heat and momentum transfer properties.

### 2.1 Barotropic instabilities of shear layers and jets

#### 2.1.1 CSP stability criteria and energy exchanges

For a purely barotropic flow, the CSP analysis reduces $\partial Q/\partial y$ to $\beta - \partial^2 U/\partial y^2$, leading to stability criterion $i$ taking the familiar Rayleigh-Kuo form for the stability of a two-dimensional flow. In effect this represents the need for an inflection point (or related feature) in a zonal shear flow as a necessary but insufficient criterion for its instability. Such a criterion is necessary to enable the possibility of supporting a pair of zonally-propagating Rossby-like waves that can phase-lock across the inflection point and lead to the growth of an instability e.g. via over-reflection (e.g. see Lindzen and Hou, 1988). The combination of local vorticity gradients and Doppler shifting of Rossby-like wave trains located in different parts of the flow allow for the possibility of pairs of such wave trains to propagate at the same velocity relative to their common frame of reference. Provided their lateral extent is sufficient for them to interact, these pairs of wave trains can remain mutually coherent and stationary in a common frame of reference moving with the waves, allowing them to interact strongly and draw kinetic energy from the background flow if their phase relationship is favourable.

This generally entails a phase tilt across the channel in such a way that the waves "lean into" the sheared zonal flow so that the momentum flux $\overline{u^*v^*}$ acts to reduce the shear of the background zonal flow. This correlation between the eddy momentum flux and zonal shear is quantified by the CK term in the Lorenz energy budget:

$$\mathrm{CK} = \left\langle \frac{\mathrm{d}\overline{[u]}}{\mathrm{d}y}\overline{[u^*v^*]} \right\rangle, \tag{2}$$




where angle brackets denote a vertically integrated areal average and square brackets denote a time average. This leads naturally to CK > 0 such that kinetic energy is transferred from KZ to KE.

### 2.1.2 Barotropically unstable flows in the laboratory

It turns out that generating a flow in the laboratory with such an inflection point favourable for supporting phase-locked Rossby wave pairs is reasonably straightforward, because such a flow structure occurs spontaneously in a viscous Stewartson shear layer. A detached shear layer may be produced in a rotating, homogeneous fluid by use of differentially-rotating horizontal boundaries, e.g. by the use of a differentially rotating disk or ring at the centre of a cylindrical tank. This was originally investigated by Hide and Titman (1967) and later explored in experiments by Früh and Read (1999a), Früh and Read (1999b) and Aguiar et al. (2010). A schematic diagram of the apparatus is shown in Figure 2(a-b) and a typical apparatus illustrated in Figure 2(c).

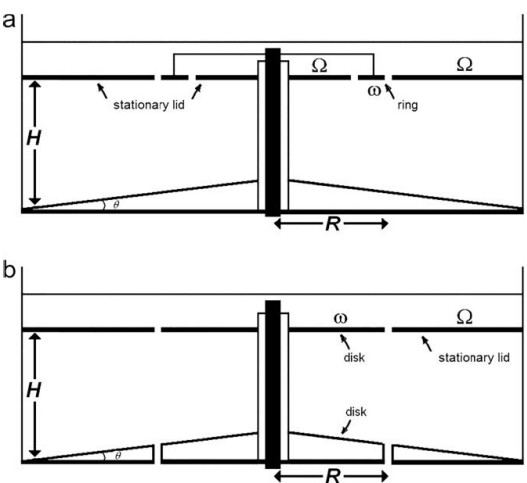

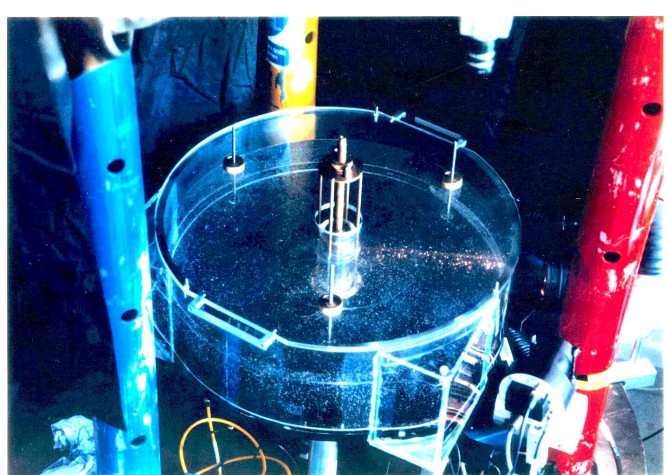

**Figure 2.** Schematic cross-sections of two versions of apparatus for producing and studying barotropic instabilities in a cylindrical tank of fluid on a rotating table at angular velocity $\Omega$. In (a) a differentially-rotating, axisymmetric ring is used to drive a barotropic jet at relative angular velocity $\omega$ at radius $R$, while in (b) a pair of differentially rotating disks, centred on the rotation axis, generate a detached Stewartson shear layer, also at radius $R$. In these cases, a topographic $\beta$-effect is also included by use of a conically sloping bottom with slope angle $\theta$. (c) Photograph of the detached shear layer apparatus as used by Früh and Read (1999a) for investigating fully developed barotropic instabilities.

The resulting directly forced zonal flow has a clear inflection point in its radial velocity profile in such a way that the radial vorticity gradient clearly changes sign. A typical example is illustrated in Figure 3 for a jet-like flow, driven by a differentially-rotating ring. The azimuthal velocity is shown in Fig. 3a while the corresponding vorticity gradient is shown in Fig. 3b.

This would appear to suggest that the flow is unstable, yet is observed in this state in a time average under conditions in which a regular azimuthal wavenumber $m = 6$ flow is found. The CSP condition, however, is only a necessary condition for instability, not a sufficient one. An alternative condition for instability was considered by Arnol'd (1966) and is often

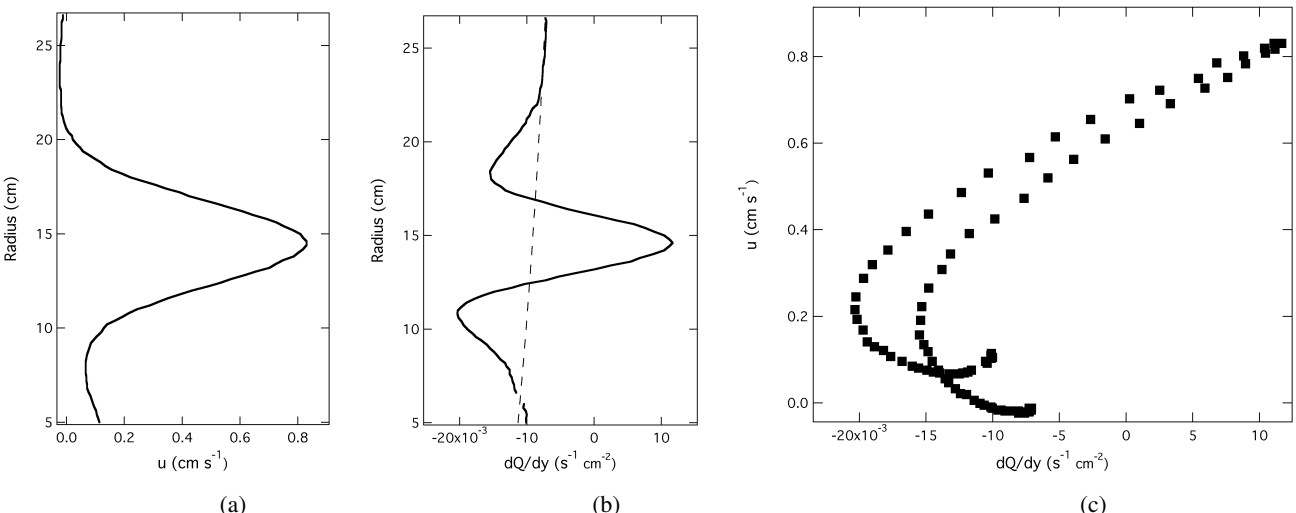

(a)  (b)  (c)

**Figure 3.** A typical azimuthal and time mean jet flow, driven by a differentially rotating ring as used by Aguiar et al. (2010), derived from particle image velocimetry measurements. (a) shows the radial profile of azimuthal velocity while (b) shows the corresponding radial gradient of shallow water potential vorticity, clearly confirming a change of sign of $\partial Q/\partial y = -\partial Q/\partial r$ at $r \simeq 12$ and 17 cm, on either side of the barotropic jet ($\partial Q/\partial y$ due solely to the gradient of $2\Omega/h$ is shown as a dashed line). (c) shows a scatter plot of $u$ against $\partial Q/\partial y$, illustrating the nearly linear relationship between the two quantities within the barotropically unstable jet for $\overline{u} > 0.2$ cm s$^{-1}$.

referenced as "Arnol'd's second stability theorem" (e.g. Dowling, 1995). This approach considers the instability that could
arise from the interaction of two (or more) parallel trains of Rossby waves, each centred on a local maximum or minimum
of potential vorticity gradient. The existence or otherwise of a steadily moving reference frame in which both wave trains,
Doppler shifted by the zonal flows in their vicinity, can be at rest relative to each other, provides a criterion for their sustained
mutual interaction and hence exchange of momentum to gain energy at the expense of the background zonal flow. The resulting
condition for perturbations to be stable is of the form:

$$\left( \frac{\overline{u} - \alpha}{\partial \overline{Q}/\partial y} \right) \geq L_0^2, \tag{3}$$

where $L_0$ is a constant length scale which, depending on the model, may represent the Rossby radius of deformation ($L_D$), and
$\alpha$ is a constant representing the speed of the waves in a shifted inertial reference frame. The marginal stability condition is set
by replacing the inequality in Eq (3) by an equality, such that $\overline{u}$ is linearly related to $\partial \overline{Q}/\partial y$ with an offset $\alpha$ at the intercept
where $\partial \overline{Q}/\partial y = 0$ (e.g. Dowling, 1995).

In practice, however, marginally stable real flows with both forcing and dissipation will not obey this condition exactly,
but may approximate to it if the forcing and dissipation are relatively weak compared to the advective transport of $Q$. For
the laboratory flow presented in Fig. 3(a) and (b), this is illustrated in Fig. 3(c), which shows a plot of $\overline{u}$ against $-\partial \overline{Q}/\partial r$




(= $\partial Q/\partial y$ with $y$ directed towards the rotation axis), where $Q$ is now the shallow water potential vorticity

$$Q = \frac{\zeta + 2\Omega}{h(r)}, \tag{4}$$

$\zeta$ is the relative vorticity and $h(r)$ is the radius-dependent depth of the fluid layer. Fig. 3(c) clearly shows a strong and nearly linear correlation between $\partial Q/\partial y$ and $\overline{u}$ where $\overline{u} > 0.2$ cm s$^{-1}$, corresponding to the core of the zonal jet in Fig. 3(a). Thus, even though the equilibrated time-averaged flow continues to satisfy the CSP condition for instability, it has effectively stabilised the flow to a configuration that is close to marginal stability with regard to Arnol'd's second stability condition (hereafter *Arnol'd II*), as a result of a process which can be regarded as a form of *barotropic adjustment*.

### 2.1.3 Circulation regimes and wavenumber selection

The resulting flows thus equilibrate to a marginally stable state in which the original jet meanders in the form of a train of azimuthally travelling waves, on either side of which are patterns of closed vortices whose vorticity matches in sign with that of the flanks of the zonal jet. The flow is typically found to select just one dominant zonal wavenumber in many of the published experiments (e.g. see Figure 4), the wavelength of which depends mainly upon a combination of the Rossby $Ro$ and Ekman $E$

numbers, commonly defined (e.g. Früh and Read, 1999a, b) as

$$Ro = \frac{R\omega}{2\overline{\Omega}d}, \tag{5}$$

$$E = \frac{\nu}{\overline{\Omega}d^2}, \tag{6}$$

where $R$ is the radius and $\omega$ the relative angular velocity of the shear layer or jet, $\nu$ the kinematic viscosity of the fluid and $d$ the depth of the fluid layer. $\overline{\Omega}$ is the mean angular velocity of the fluid, representing the average of the turntable rotation rate $\Omega$

and the differential rotation rate $\omega$.

As shown by Niino and Misawa (1984) and Früh and Read (1999a), although the inviscid CSP criterion $i$ or (more realistically) Arnol'd II criterion needs to be satisfied for the driven jet to be unstable, the observed stability boundary in a real, viscous fluid is also consistent with the existence of a critical Reynolds number $Re_c$, defined with respect to a length scale of the same order as that of the $E^{1/4}$ Stewartson layer, such that

$$Re_c = \frac{UL_S}{\nu} \simeq Ro.E^{-3/4} \simeq 27, \tag{7}$$

where

$$L_S = (E/4)^{1/4}d \tag{8}$$

is the width of the viscous Stewartson layer. This expression provides a reasonably accurate description of the observed stability boundary (e.g. see Fig. 4) for both the jet and detached shear layer flows, such that for $Re < Re_c$ the flow observed is essentially

axisymmetric with no discernible wave perturbation.

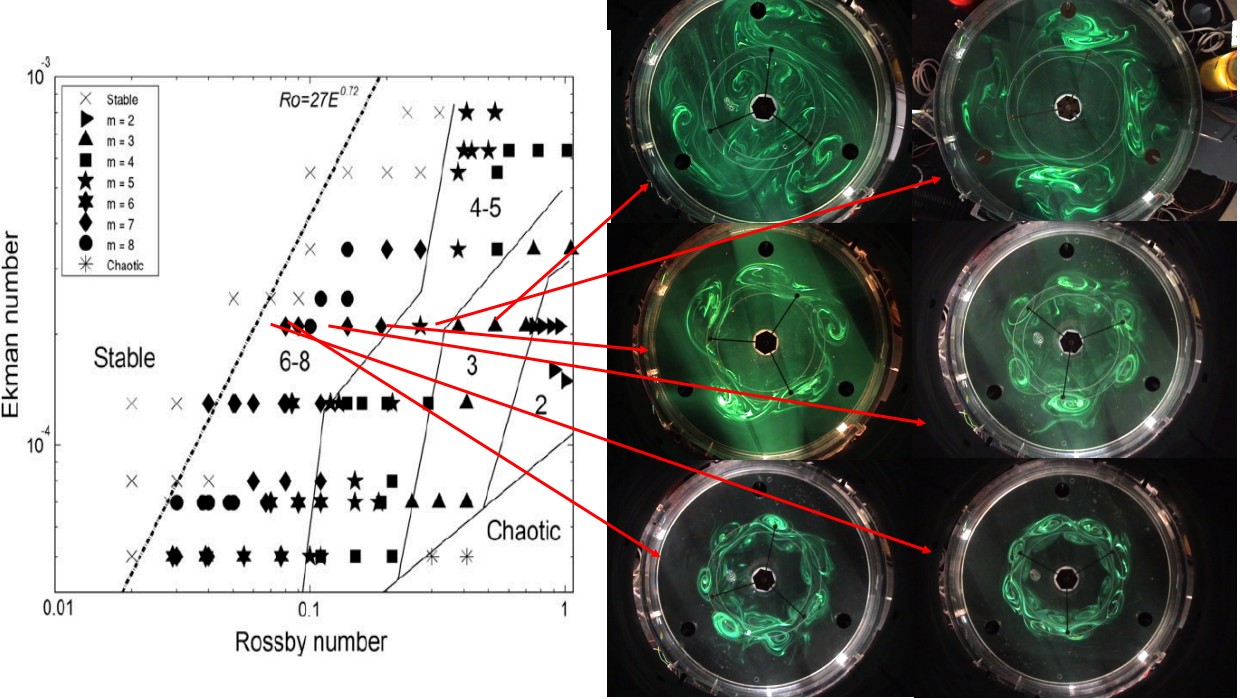

**Figure 4.** Regime diagram for the barotropic jet flows as a function of Rossby and Ekman number, as obtained by Aguiar et al. (2010), showing snapshots of typical equilibrated flows as visualised using dye and side illumination. The trend towards higher wavenumber as the stability boundary at $Re = Re_c$ is approached is clearly seen.

The corresponding boundaries between different flow regimes dominated by different wavenumbers are observed empirically to lie almost parallel to the main stability boundary, suggesting that the wavenumber selected is largely determined by the effective Reynolds number, and hence the effective supercriticality of the barotropic instability ($Re - Re_c$). In fully developed flows this may be consistent with other work on scale selection in barotropic instability, which associates the most unstable

wavelength with a scale comparable to the width of the background jet or shear layer (e.g. Sommeria et al., 1991; Vallis, 2017). In these barotropic instability experiments, however, the equilibrated amplitude of unstable waves grows with supercriticality, effectively broadening the width of the original jet beyond its initial Stewartson layer width. At its equilibrated finite amplitude, therefore, strongly supercritical flows will tend to favour longer wavelength disturbances, qualitatively consistent with the observed ordering of different wavenumber regimes. This can be clearly seen in Fig. 4, in which the lower wavenumber

states are seen to fill the entire tank with large meanders and vortices, wheres high wavenumber flows tend to be more tightly confined in radius to the vicinity of the unstable jet beneath the rotating ring. At large supercritical Reynolds numbers, however, the equilibrated flow may no longer be steady but may become chaotic, as observed at the lowest Ekman numbers (see Fig. 4), although this has not been explored in much detail as yet.




This method of forcing and maintaining a barotropically unstable flow is relatively strong and fast, so even at large equi-
librated amplitudes the zonally averaged flow maintains a reversal in the lateral gradient of potential vorticity. The forcing
therefore would seem to be too strong to allow the zonally symmetric component of the flow to maintain itself close to a
marginally unstable state. The resulting equilibrated flow is therefore strongly nonlinear, especially at relatively large effective
Reynolds numbers.

### 2.2 Baroclinic instabilities

Baroclinic instabilities require the maintenance of a zonally symmetric distribution of density or temperature that is statically
stable in the vertical direction together with a horizontal temperature gradient. This configuration has been studied intensively
for more than 60 years in the laboratory using the thermally-driven, rotating annulus (see Figure 5). A fluid is contained between
two upright, coaxial metal cylinders, which can be maintained at two different temperatures. Most typically, the inner cylinder
is maintained at a cooler temperature than the outer, schematically representing the thermal contrast maintained between the
equator and poles of an Earth-like planetary atmosphere. The annular tank is mounted on a rotating table which rotates at
angular velocity $\Omega$, with the axis of rotation aligned with the axis of symmetry of the cylinders, again by analogy with a
planetary atmosphere. The thermal contrast, $\Delta T$, between the cylinders drives an axisymmetric overturning circulation in the
annular channel, mainly confined to boundary layers, which enables a stable stratification to develop and equilibrate, while the
effect of background rotation is to induce a vertically sheared azimuthal flow as fluid moves between inner and outer cylinders
while roughly conserving its angular momentum.

At a fast enough rotation rate, radial flow becomes largely confined to shallow Ekman layers close to the horizontal (ther-
mally insulating) boundaries, while the isotherms in the interior of the annular channel develop an inclined slope with respect
to the horizontal, thus effectively storing potential energy. The flow patterns observed in such a system are then found to depend
strongly upon both the rotation rate and the imposed thermal contrast between the cylindrical boundaries (e.g. see Hide and
Mason, 1975; Read et al., 2015, for reviews). The approximate sequence of bifurcations observed between different clrculation
regimes is shown schematically in Figure 6 as a function of the principal dimensionless thermal Rossby number, $\mathrm{Ro}_T$, and
Taylor number, $\mathcal{T}$, defined respectively as

$$\mathrm{Ro}_T \quad = \quad \frac{g\alpha\Delta T d}{\Omega^2 L^2}, \tag{9}$$

$$\mathcal{T} \quad = \quad \frac{4\Omega^2 L^5}{\nu^2 d}, \tag{10}$$

where $g$ is the acceleration due to gravity, $\alpha$ the volumetric expansion coefficient of the fluid, $\nu$ is the kinematic viscosity, $L$ is
the horizontal width of the annular channel and $d$ its depth.

As is well known, for a given imposed thermal contrast $\Delta T$, the flow is observed to be stable to baroclinic instabilities for
rotation rates slower than a critical value (corresponding to $\mathrm{Ro}_T \gtrsim 2$), resulting in an axisymmetric overturning circulation with
prograde flow at upper levels. For higher rotation rates ($\mathrm{Ro}_T \lesssim 2$), the flow becomes unstable to baroclinic disturbances that
cause the originally axisymmetric azimuthal flow to meander in the form of a train of azimuthally propagating waves. These




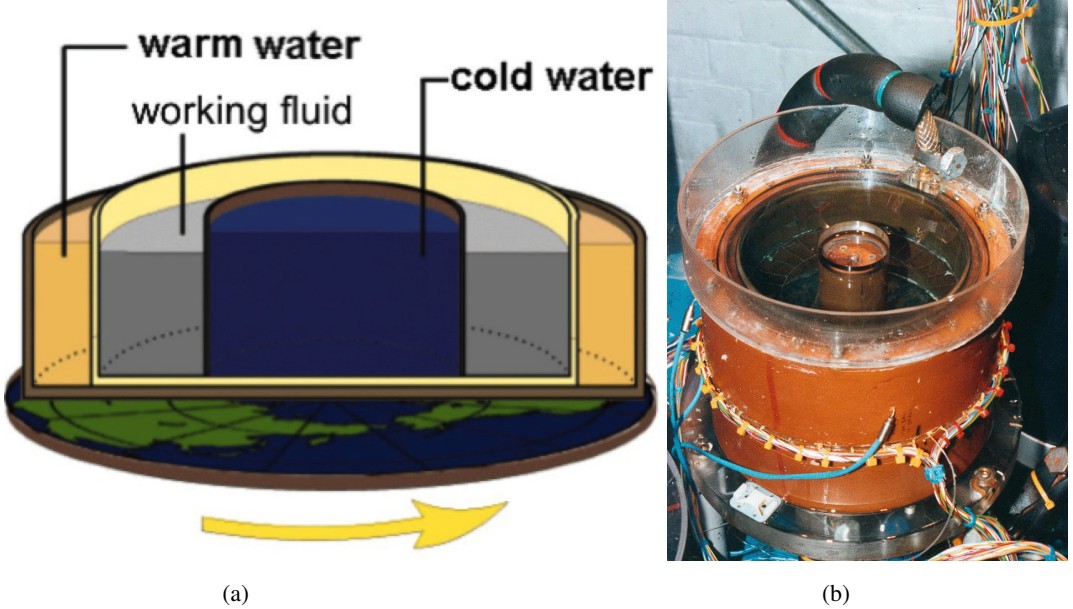

(a)           (b)

**Figure 5.** Schematic layout of the thermally-driven, rotating annulus experiment (a) as a vehicle for the study of rotating, stratified flows, with (b) a photograph of a typical apparatus in the laboratory, clearly showing the metallic inner and outer cylinders and connections to heat exchangers for maintaining the temperature of the outer wall and a ring of thermocouple junctions to measure temperatures in the interior.

baroclinic waves are the fully developed manifestation of baroclinic instability, sometimes referred to as *sloping convection*. (e.g. Hide, 1969; Hide and Mason, 1975), and may take the form of either regular, near-monochromatic wave trains or more chaotic or even turbulent flows at the highest rotation rates.

### 2.2.1 CSP instability criteria

Although from an energetic viewpoint it is reasonably clear why the basic state maintained by the differential heating in the rotating annulus experiment can lead to the release of potential energy to energise the growth of wave-like perturbations, it is less clear how the flow might satisfy the CSP criteria for instability. Early work by Hide (1969) compared the conditions under which active sloping convection was observed in the laboratory with various extensions of the linear model of baroclinic instability by Eady (1949). Provided account was taken of the dissipative effects of viscous Ekman layers adjacent to the upper 240   and lower boundaries, these comparisons found generally good quantitative agreement for the onset of the instability around a Burger number of order $\mathrm{Ro}_T/4 \simeq 0.5$. Such an analogy, however, presumes that potential vorticity gradients in the interior are negligible compared with the thermal gradients at the horizontal boundaries, satisfying the CSP stability condition through criterion *iv*.

    In practice, however, the strong radial flow within the Ekman layers in typical annulus experiments rapidly transfers hot or 245   cold fluid across the domain, maintaining relatively weak horizontal thermal gradients at the top of each Ekman layer. This





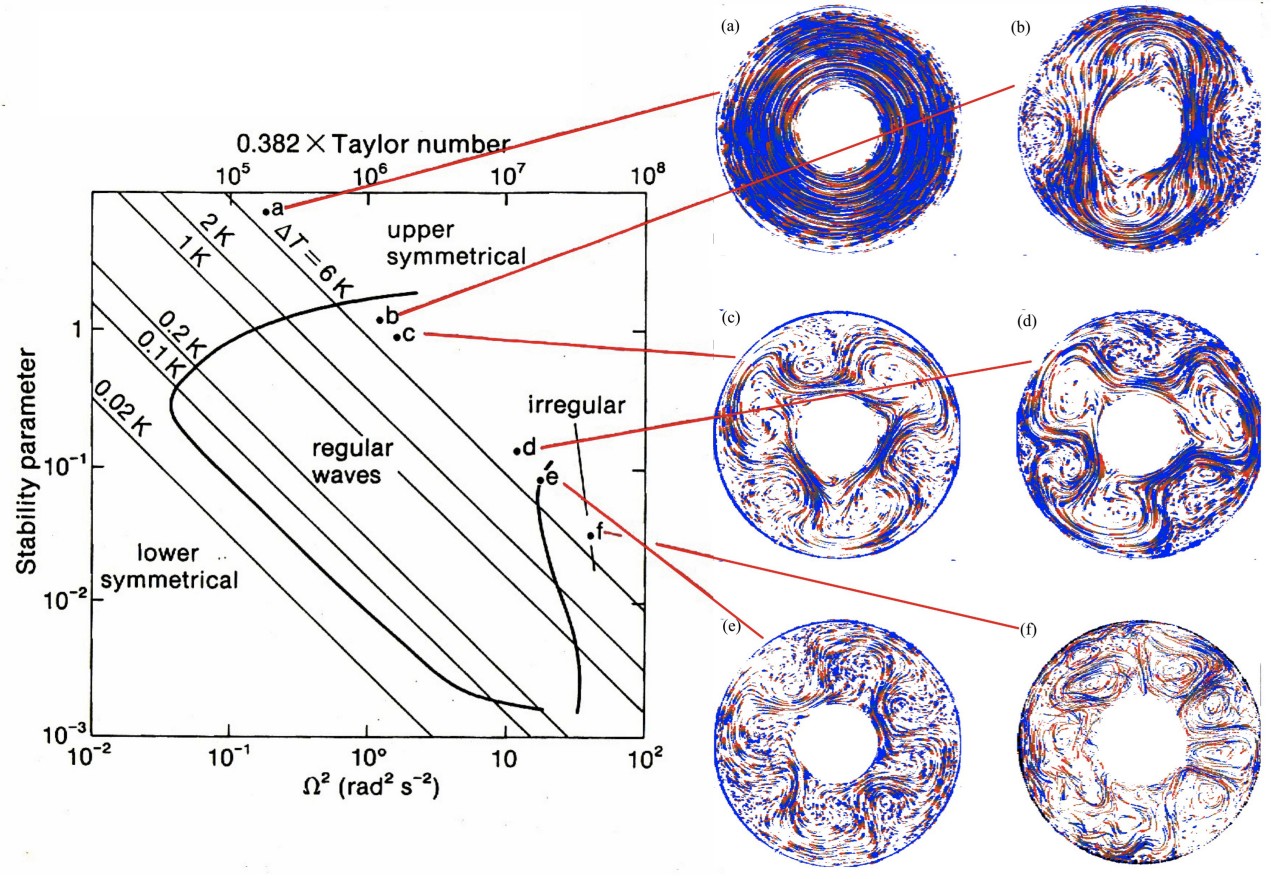

**Figure 6.** Regime diagram for the baroclinically stable and unstable flows in a thermally-driven, rotating annulus as a function of thermal Rossby and Taylor number (e.g. Read et al., 2015), showing snapshots of typical equilibrated flows as visualised using neutrally buoyant tracer particles and side illumination to obtain streak images.

would suggest that the potential vorticity dynamics is controlled much less by the structure of the flow close to the horizontal boundaries and is more strongly influenced by a change of sign of $\partial Q/\partial y$ in the interior flow. This would indicate that the basic zonal flow maintained by the differential heating at the side boundaries is closer in character to an internal jet (e.g. Charney and Stern, 1962; Bell and White, 1988). Baroclinic or barotropic instability would therefore occur through satisfying the CSP

conditions directly through criterion *i*. The consequences of this in the context of rotating annulus experiments were explored by Bell and White (1988), who noted that idealised, inviscid, baroclinic internal jets without any lateral shear would become unstable at around the same value of Burger or thermal Rossby number as (or slightly larger than) the classical Eady problem. This was broadly in agreement with experiments, which did show a tendency for the onset of instability to occur at slightly larger values of $\mathrm{Ro}_T$ than the Eady problem would suggest. However, the critical value of $\mathcal{B}u$ or $\mathcal{R}o_T$ was found to be quite

strongly sensitive to the addition of lateral shear to a baroclinic internal jet (Bell and White, 1988).





Numerical analyses using more realistic linearised models based on the full Boussinesq Navier-Stokes equations with molecular viscosity and thermal diffusion (e.g. Lewis and Nagata, 2004; Lewis et al., 2015, and references therein), however, show close agreement with experimental measurements and also reveal sensitivity of the instability to other factors such as the Prandtl number, especially at low values of both $\mathcal{R}o_T$ and $\mathcal{T}$ along the boundary of the so-called "lower symmetric" regime.

At high Prandtl number, the boundary roughly follows a line of constant $\Delta T$, indicative of a critical Rayleigh or Grashof number ($\mathcal{G}r_c$) in a similar way to the criterion for Rayleigh-Bénard convection. The product $\mathcal{R}o_T.\mathcal{T}$ actually has the form of a Grashof number, such that $\mathcal{R}o_T.\mathcal{T} = \mathcal{G}r_c$ is consistent with the shape of the lower left instability boundary shown in Fig. 6.

### 2.2.2 Lorenz energy cycle

As instabilities develop and equilibrate, azimuthally travelling waves emerge with characteristic three-dimensional structures
and azimuthal phase tilts in radius and height. These result in eddy fluxes of heat and momentum that lead to energy conversions between potential and kinetic energies in the zonally averaged and perturbation fields. A typical example of the Lorenz energy cycle for an equlibrated baroclinic wave developed from a baroclinically unstable initial state is shown in Figure 7. This was computed by Young (2014) from a numerical simulation of flow in a rotating annulus experiment from solutions of the Boussinesq Navier-Stokes equations in cylindrical annular geometry. The boundary conditions represent non-slip, rigid
sidewalls and horizontal endwalls, thermally insulating endwalls and isothermal sidewalls at temperatures $T_a$ and $T_b$ at inner and outer radii $r_a$ and $r_b$, with $\Delta T = T_b - T_a = 4$ K, $r_a = 2.5$ cm, $r_b = 8.0$ cm, $d = 14$ cm and rotation rate $\Omega = 3.0$ rad s$^{-1}$, corresponding to $\mathcal{R}o_T = 0.056$.

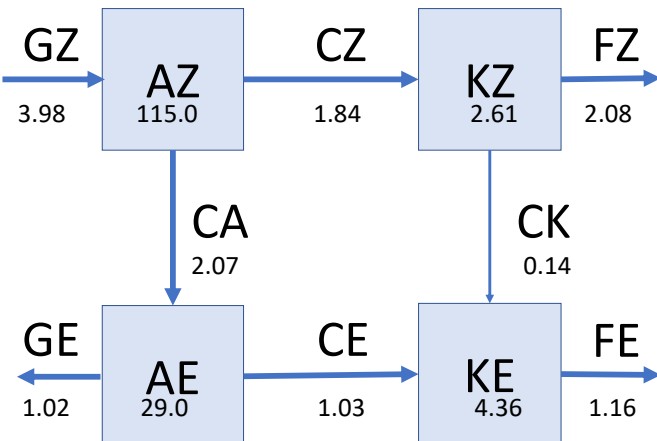

**Figure 7.** An example Lorenz energy cycle, computed by Young (2014) from a numerical simulation of an equilibrated baroclinic wave flow in a differentially-heated, rotating annulus. Energy reservoir and conversion and source/sink terms are in units of $10^{-7}$ J kg$^{-1}$ and W kg$^{-1}$ respectively.





The values shown in Fig. 7 clearly indicate that the conversion terms implicated in energising baroclinic instability (CA and CE; see Fig. 1(b)) are dominant in the exchanges between eddies and the zonal mean flow. CK is at least a factor of 10 smaller

than CA though in the same sense for the generation of KE, indicating that the dominant instability has a somewhat mixed baroclinic-barotropic character. The direct zonal mean exchanges (CZ) are also of a similar magnitude to CA and CE, and are consistent with a thermally direct overturning circulation in parallel with the conversion of available potential energy to KE.

### 2.2.3  Signatures of baroclinic adjustment in heat transport?

As mentioned above, baroclinic adjustment is commonly associated with a tendency for baroclinic instability to equilibrate at
finite amplitude in such a way as to modify the initially unstable basic state towards a less unstable configuration. But it is less clear how this tendency might manifest itself in practice. However, one possible way this might be observable could be in the way heat transport by baroclinic disturbances varies with external parameters as the instability threshold is exceeded.

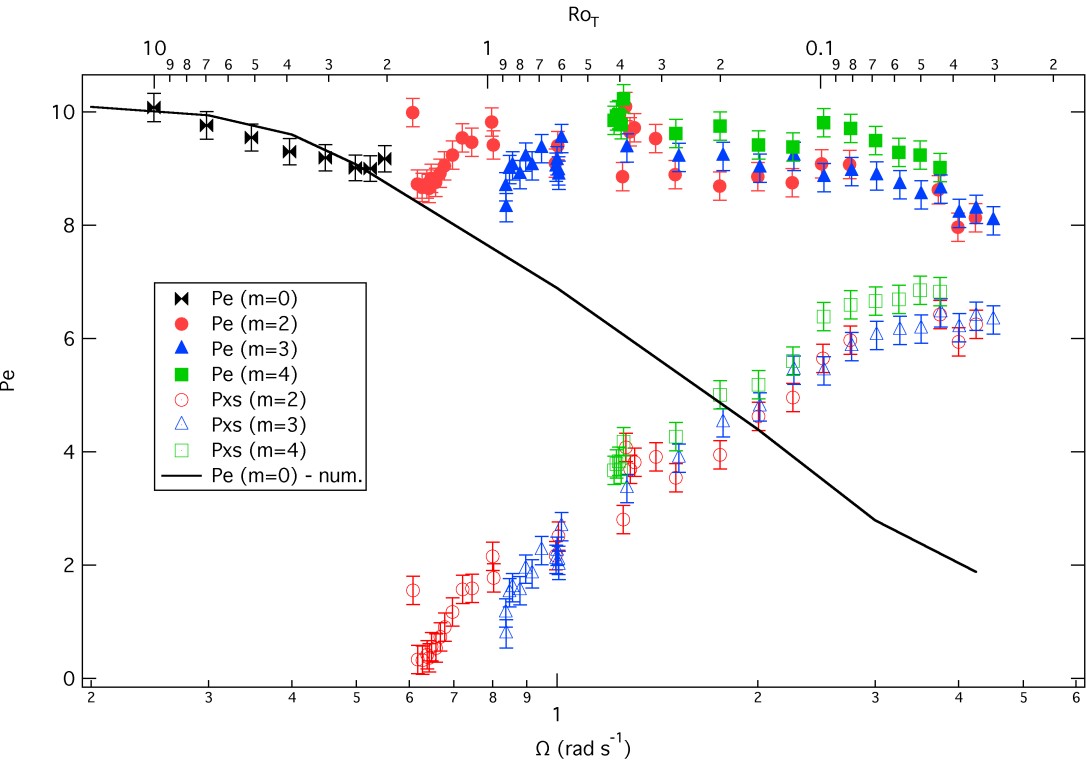

**Figure 8.** Variations of radial heat transport by baroclinic eddies and axisymmetric flows in a differentially heated rotating annulus, as measured in the laboratory (Read, 2003) as a function of rotation rate and thermal Rossby number $Ro_T$. Heat transport is indicated by the dimensionless Nusselt number Nu. Solid symbols represent total heat transport as measured by calorimetry, while the continuous line represents the total heat transport by the axisymmetric flow alone. Open symbols indicate the different in Nu between measured total heat transport and the axisymmetric component, representing the heat transport due to eddies alone.





Figure 8 shows a series of calorimetric measurements of total heat transport in a differentially heating rotating annulus experiment for fixed $\Delta T$ as $\Omega$ is varied (see Read, 2003). In this case, heat transport is non-dimensionalised with respect to

thermal conduction by the Nusselt number, Nu, defined by

$$\mathrm{Nu} = \frac{\mathcal{H}\ln(r_b/r_a)}{2\pi\kappa d\rho c_H \Delta T}, \tag{11}$$

where $\mathcal{H}$ is the total heat transport in W, $\kappa$ is the thermal diffusivity of the working fluid and $c_H$ its specific heat capacity. Thus the solid symbols in Fig. 8 show little variation with $\Omega$ or $\mathcal{R}o_T$ until the highest rotation rates. This corresponds to where the regular wave regime in the laboratory experiments begins to break down towards small-scale irregular flows and geostrophic

turbulence. The solid line in Fig. 8, on the other hand, shows the variation of Nu for a purely axisymmetric flow, obtained from numerical simulations in which non-axisymmetric eddies were suppressed (see Read, 2003, for details). This shows that the axisymmetric flow under similar conditions decays much more quickly with $\Omega$ than when baroclinic waves are allowed to develop. The corresponding contribution to the dimensionless total heat transport by baroclinic waves is indicated by the open symbols, which are obtained from the difference between total heat transport and that of the pure axisymmetric flow. This

clearly shows that the eddy contribution increases systematically from zero as the rotation rate is increased beyond the critical value for the onset of instability, until it begins to saturate and even begin to turn over for $\mathcal{R}o_T < 0.1$.

The observation that total heat transport may be almost independent of rotation rate over a wide range of $\Omega$, from the axisymmetric regime into the fully developed baroclinic wave regime, has been noted since the early work of Bowden (1961), who was the first to carry out calorimetric measurements of heat transport in rotating annulus experiments. This tendency for the

heat transport to remain close to its value at $\Omega = 0$, even though the (mainly axisymmetric) boundary layer contribution should reduce substantially over the same range of parameters, is unlikely to be coincidental, but more likely reflects a systematic result of equilibrated baroclinic instability in modifying the overall flow to transport almost as much heat (and therefore release almost as much potential energy) as the non-rotating flow. The non-rotating flow represents a fully relaxed state, in which isotherms and isopycnals are as near horizontal as possible (except in conductive boundary layers), and is therefore a state that

would be energetically stable to further baroclinic instabilities. In this sense, these results demonstrate that rotating annulus flows are capable of exhibiting a relatively strong form of baroclinic adjustment throughout the regular baroclinic wave regime. It is only when the regular regime begins itself to become unstable, and the dominant energetic length scale of the baroclinic waves becomes significantly smaller than the width of the imposed baroclinic zone, that the eddies are no longer able to sustain the level of heat transfer required to maintain the fully relaxed flow state.

Such a tendency would seem to be quite general, and so it is of interest to see whether similar trends might be observable in a planetary atmosphere, in which differential heating is maintained by (radiative) relaxation towards a baroclinically unstable radiative equilibrium state.





## 3 Baroclinic and barotropic instabilities in idealised planetary atmospheric circulations

The Earth and other Solar System planets present us with just a few samples of atmospheric circulation systems in different
parts of a broader parameter space (e.g. see Read, 2011). A better way of exploring trends and scaling of various properties of planetary atmosphere circulations is to use a simplified numerical model, in which the key forcing processes are represented schematically but in a physically consistent manner as a function of the principal planetary parameters, so that the model can be used to explore the relevant dynamical response. This approach has a long history, dating back to the early work of Hunt (1979), Williams and Holloway (1982) and Geisler et al. (1983), but has become more common in recent years, inspired in part
by the diversity of newly discovered extrasolar planets (e.g. Merlis and Schneider, 2010; Mitchell and Vallis, 2010; Kaspi and Showman, 2015; Wang et al., 2018). Here we focus on diagnostics of energetics, instabilities and heat transfer in a typical set of idealised GCM simulations, based on the work of Wang et al. (2018).

### 3.1 Circulation regimes

As demonstrated by various authors (e.g. Kaspi and Showman, 2015; Wang et al., 2018), many generic features of the large-
scale circulation of almost any planetary atmosphere can be captured in a general circulation numerical model that solves the hydrostatic primitive equations of meteorology, together with simple parameterizations of diabatic heating and small-scale turbulent mixing and friction. In recent work, Wang et al. (2018) has used the University of Hamburg PUMA model (e.g. Frisius et al., 1998; Fraedrich et al., 2005) to explore the dependence of the simulated fully three-dimensional, time-dependent circulation of an Earth-like planetary atmosphere on dimensionless parameters such as the thermal Rossby number, $Ro_T$,
as quantities such as the planetary rotation rate, $\Omega^* = \Omega/\Omega_E$ (where $\Omega_E$ is the rotation rate of the Earth), are varied. This model represents fields in the horizontal as projections onto sets of spherical harmonic functions but in finite difference form in the vertical. Diabatic heating and cooling is represented by a linear relaxation towards a prescribed zonally-symmetric temperature field, intended to represent the diurnally and seasonally averaged radiative-convective equilibrium of an Earth-like planet, with a prescribed relaxation timescale $\tau_R$. Surface friction is represented by a height-dependent, linear Rayleigh friction
parameterization with local timescale $\tau_F$ and total (Ekman) spin-down timescale of $\tau_S$.

   The model was run to equilibrium over 10-20 Earth years before computing various diagnostics of the statistically equilibrated state. The circulation regime could then be characterised as a function of various dimensionless parameters, such as the thermal Rossby number,

$$\mathcal{R}o_T = \frac{R\Delta\theta_{EP}}{\Omega^2 a^2},$$ (12)

where $R$ is the specific gas constant, $\Delta\theta_{EP}$ is the horizontal temperature contrast between equator and poles and $a$ is the planetary radius, the Burger number,

$$\mathcal{B}u = \frac{N^2 H^2}{4\Omega^2 a^2} \simeq \frac{R\Delta\theta_z}{4\Omega^2 a^2},$$ (13)





where $\Delta\theta_z$ is the vertical contrast in potential temperature, and frictional or radiative Taylor numbers,

$$\mathcal{T}_{(F,R)} = 4\left(\Omega(\tau_F, \tau_R)\right)^4. \tag{14}$$

The results of a scan through parameter space as $\Omega^*$ is varied may be summarised in a regime diagram with respect to $\mathcal{R}o_T$ and $\mathcal{T}_F$, which identifies and classifies different circulation regimes according to their location in parameter space.

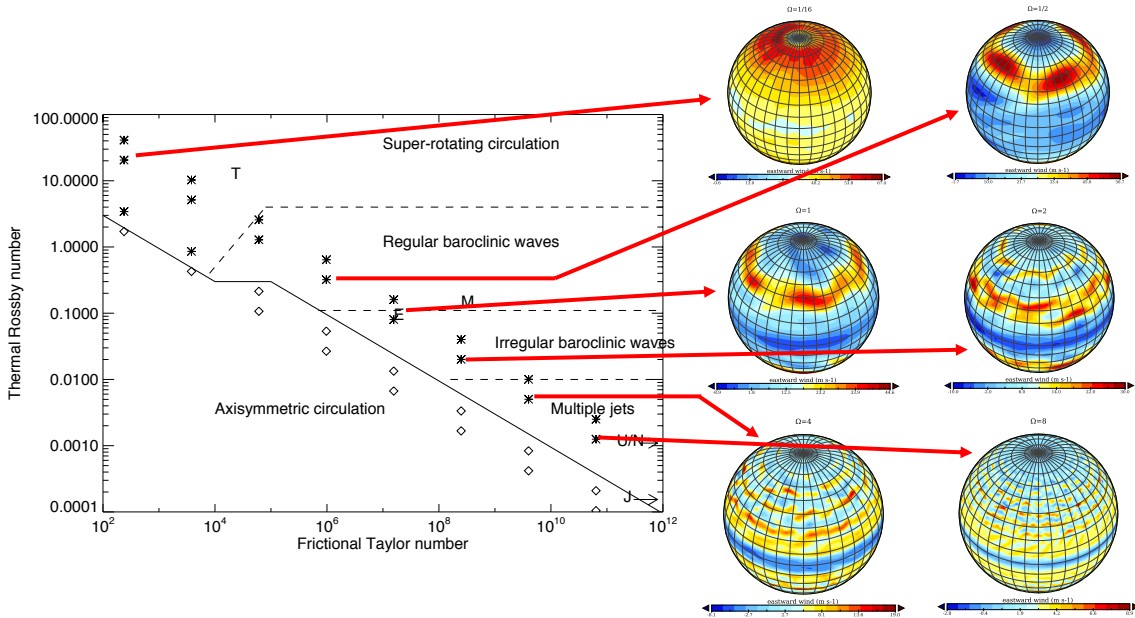

**Figure 9.** Regime diagram showing the various circulation regimes obtained in a simplified atmospheric general circulation model (Wang et al., 2018) with respect to characteristic dimensionless parameters ($\mathrm{Ro}_T$ and $\mathcal{T}_F$). Stars refer to experiments in which wavy flows are observed, whereas open diamonds indicate experiments in which axisymmetric flows were found. The approximate location in parameter space of some Solar System planets (Earth, Mars, Titan, Jupiter, Saturn, Uranus and Neptune) are labeled by their initial letters. The solid line delineates the boundary between axisymmetric circulations and circulations with wavy/turbulent flows. The dashed lines indicate the boundaries between different circulation regimes within the wavy/turbulent region. Some example snapshots of typical circulation patterns, visualised by shading of the zonal wind speed at the 200 hPa level, are indicated by the arrows.

Figure 9 presents the regime diagram for the PUMA simulations of Wang et al. (2018), together with some sample snapshots of typical flows at different planetary rotation rates which present visualisations of the zonal wind speed at the upper tropospheric level of 200 hPa. As discussed more fully by Wang et al. (2018), the sequence of circulation patterns follows a

clear trend with planetary rotation rate. Keeping $\Delta\theta_E P$ constant, the locus of varying rotation rate follows a diagonal path from upper left to bottom right in $(\log\mathcal{R}o_T, \log\mathcal{T}_F)$. At the lowest values of $\Omega^*$ the circulation is dominated by unsteady cir-





| $\Omega^*$ | $\mathcal{R}o_T$ | $\mathcal{T}_F$ | $\mathcal{T}_R$ |
|---|---|---|---|
| 1/16 | 20.5 | 238 | $1.71 \times 10^5$ |
| 1/8 | 5.14 | $3.80 \times 10^3$ | $2.73 \times 10^6$ |
| 1/4 | 1.28 | $6.09 \times 10^4$ | $4.67 \times 10^7$ |
| 1/2 | 0.32 | $9.74 \times 10^5$ | $6.98 \times 10^8$ |
| 1 | 0.08 | $1.56 \times 10^7$ | $1.12 \times 10^{10}$ |
| 2 | 0.02 | $2.49 \times 10^8$ | $1.79 \times 10^{11}$ |
| 4 | 0.005 | $3.99 \times 10^9$ | $2.86 \times 10^{12}$ |
| 8 | 0.0013 | $6.38 \times 10^{10}$ | $4.58 \times 10^{13}$ |

**Table 1.** Key dimensionless parameters for the baseline set of numerical simulations with $\Delta\theta_{EP} = 60$K, $\tau_{ft} = 5$ Earth days and $\tau_R = 25.9$ Earth days, as defined by Equations (12) and (14).

cumpolar vortices, a large-scale meridional overturning and strongly super-rotating zonal winds. These evolve into a pattern of more Earth-like, meandering mid-latitude jet streams at intermediate values of $\Omega^*$. This then breaks down into a more irregular pattern of parallel zonal jets, the number of which increases with increasing $\Omega^*$.

From an examination of the temperature and PV structure of the flow, it is evident that the basic circulation in these models allow for either barotropic or baroclinic instabilities, through satisfying either CSP criteria *i* or *iii*. In the latter case, the method of diabatic forcing maintains a strong equatorward thermal gradient at the lower boundary which, for quasi-geostrophic conditions, enables CSP criterion *iii* to be satisfied. Where a strong mid-latitude or circumpolar jet is formed, however, then criterion *i* may be satisfied through lateral variation in the vorticity of the jet. Precisely which will dominate in particular cases, 360    however, is not immediately clear without a more detailed analysis.

### 3.2    Energetics

Figure 10 presents the results of an analysis of the time-averaged Lorenz energy cycles for each of a set of PUMA simulations from the work of Read et al. (2018) with fixed thermal forcing, covering a range of $\Omega^*$ from 1/16 to 8 (representing a range in Ro$_T$ from $\sim 20 - 10^{-3}$). The energy content of the main reservoirs (Fig. 10(a)) is seen to change from a mainly KZ dominated 365    regime at low $\Omega^*$ to a strongly AZ dominated regime at fast $\Omega^*$ ($\mathcal{R}o_T \ll 1$). Both the eddy components, AE and KE, remain relatively small but rise to a peak for rotation rates corresponding to $\mathcal{R}o_T \sim 0.1$ - 1. All terms except AZ then decay strongly with increasing $\Omega^*$ at the highest rotation rates.

Energy conversion rates exhibit even more complex variation with $\Omega^*$ (see Fig. 10(b)). For $\mathcal{R}o_T \gg 1$ the CZ term is strongly dominant, indicating a circulation that is energetically dominated by a thermally direct, zonal mean meridional overturning. 370    Within this range, eddies gain energy from the zonal mean components through both the barotropic CK and baroclinic CE terms, but CK is dominant over CE (and CA) for $\mathcal{R}o_T > 10$. At intermediate $\Omega^*$, the baroclinic conversions, CE and CA, rise to a peak around $\mathcal{R}o_T \sim 0.3$, while CK and CZ decrease and actually change sign for $\mathcal{R}o_T < 1$. This indicates that the





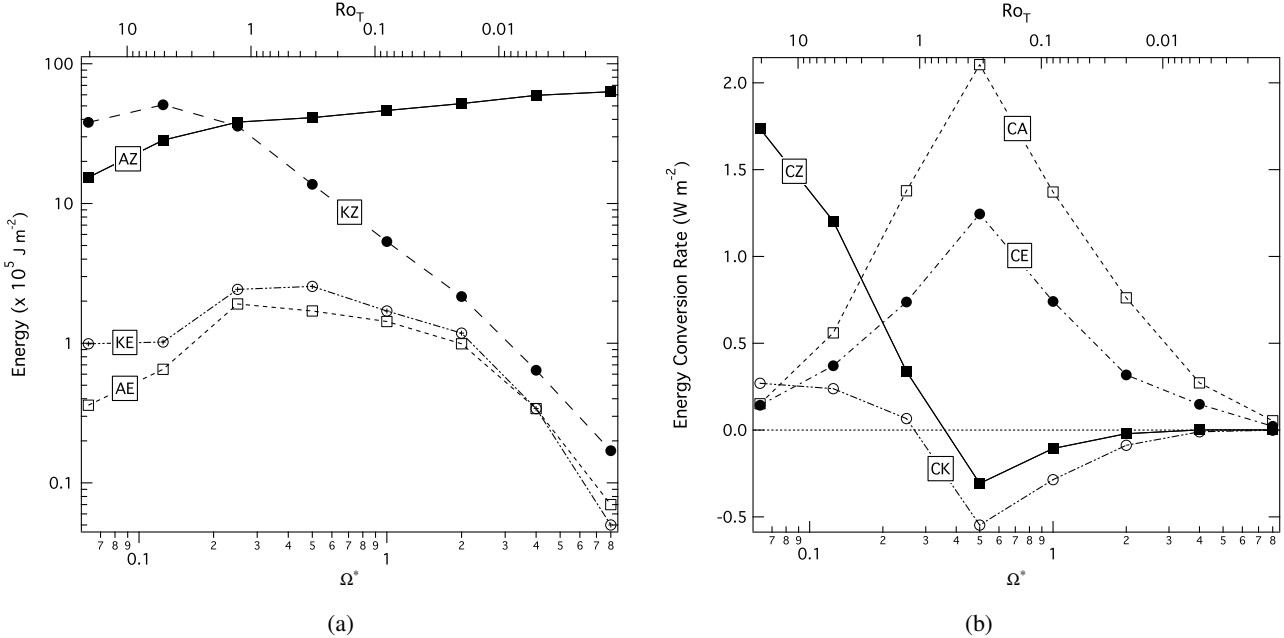

**Figure 10.** Terms in the Lorenz energy budgets for the series of PUMA-S simulations as a function of $\Omega^*$ and thermal Rossby number: (a) globally averaged energies (in $10^5 \mathrm{Jm}^{-2}$); (b) the main energy conversion rates CZ, CA, CE and CK. Conversion rates are in unit of $\mathrm{Wm}^{-2}$.

energetics of the circulation are dominated by barotropically unstable eddies at the lowest rotation rates, although the baroclinic conversion term CE is also positive in this range, indicating a mixed barotropic/baroclinic instability as the origin of these eddies. For $\mathcal{R}o_T \lesssim 1$, however, CK becomes negative while CE rises to a positive maximum around $\mathcal{R}o_T \sim 0.3$, indicating that eddies are predominantly baroclinic in character and energetics at relatively fast rotation rates. At the highest rotation rates all conversion terms are seen to decrease in magnitude as $\mathrm{Ro}_T$ decreases, leading to relatively weak and increasingly inefficient circulation.

### 3.3 Heat transfer and baroclinic adjustment

The positive contributions to both CA and CE at all rotation rates imply that the simulated circulation involves significant transfers of sensible heat across the planet. Figure 11(a) shows how the peak values of vertically integrated meridional heat transport vary with $\Omega^*$ and $\mathrm{Ro}_T$ in the PUMA-S simulations of Wang et al. (2018). The zonal mean and eddy heat transports are shown separately by open triangles and crosses respectively, with the total heat transport (zonal mean plus eddy) indicated by solid squares. This is similar in form to the results shown in Fig. 8(a) of Kaspi and Showman (2015) for their simulations using a GCM driven by a gray radiation scheme with moisture transport, apart from showing the combined total heat transport.

These results clearly show that, as in the simulations of Kaspi and Showman (2015), the zonal mean contribution to heat transport dominates at slow rotation rates (for $\mathcal{R}o_T > 1$) but decreases monotonically with increasing $\Omega^*$. The eddy contribution to heat transport, on the other hand, steadily increases (almost linearly) with $\Omega^*$ until it peaks around $\mathcal{R}o_T \sim 0.07$, beyond





which it also decreases rapidly with increasing $\Omega^*$. The sum of the two contributions, however, remains nearly independent

of $\Omega^*$ and $\mathrm{Ro}_T$ until $\mathcal{R}o_T \sim 0.07$, indicating that for values of $\mathrm{Ro}_T$ between $\sim 1$ and $0.07$ the eddy heat transport is able to compensate for the decrease in zonal mean transport to maintain the total heat transport close to its slowly rotating value.

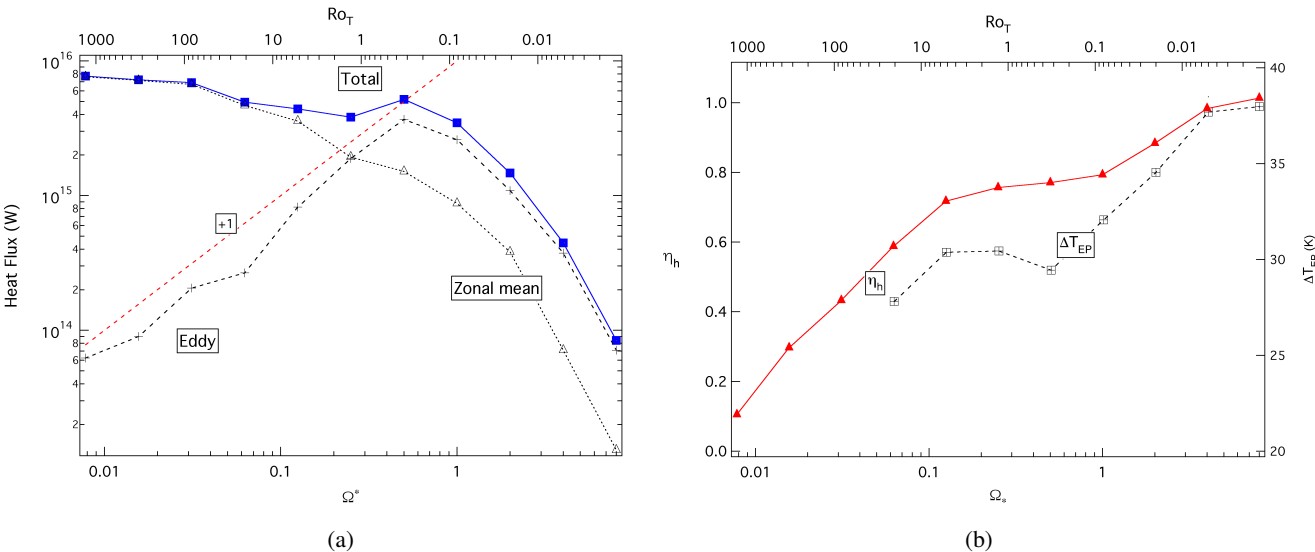

**Figure 11.** (a) Variations of peak values of meridional heat transport (in W) within the PUMA-S GCM simulations of Wang et al. (2018) and at slower values of $\Omega$, shown as total poleward transport (blue squares), zonal mean transport ($\int [c_p \overline{v}\overline{T}] 2\pi a \cos(\phi_{max}) dp/g$) and eddy transport ($\int [c_p \overline{v^* T^*}] 2\pi a \cos(\phi_{max}) dp/g$); (b) ratio $\eta$ of the slope obtained in fully active circulations in the latitude-height plane for experiments with different $\Omega^*$ to the globally mass-weighted average isentropic slope in radiative-convective equilibrium (towards which the model atmosphere relaxes, shown as filled triangles connected by solid lines), plotted alongside the variation of mean equator-pole thermal contrast in the mid-troposphere, $\Delta T_{EP}$ (shown as open squares connected by a dotted line). The (red) dashed line in (a) indicates a gradient of $+1$ with respect to $\Omega$ suggesting linear proportionality.

Fig. 11(b) indicates that the global temperature structure also reflects these variations in total heat transport as $\Omega^*$ is varied. This shows $\eta$, a measure of the mean isentropic slope in the mid-troposphere $[\partial z/\partial y]_\theta$, normalised by the mean slope of the isentropes in the radiative equilibrium state to which the thermal state of the flow is relaxed, and the mean equator-pole

temperature contrast, $\Delta T_{EP}$, also in mid-troposphere. These clearly show the isentropic slope and $\Delta T_{EP}$ increasing with $\Omega^*$, but with a plateau in this variation for $\mathrm{Ro}_T$ between $\sim 1$ and $\sim 0.1 - 0.05$.

These results show a close similarity with the heat transport results discussed in Section 2.2 for thermally-driven rotating annulus experiments, and clearly indicate a regime in which eddy heat fluxes exert a strong influence on the thermal structure of the global circulation. This is despite an absence of a clear instability threshold for baroclinic instability in the GCM

simulations, unlike what is found in the annulus experiments. However, the onset of this compensation effect of eddy heat fluxes against the decreasing zonal mean transport (effectively acting as an "eddy thermostat") around a value of $\mathcal{R}o_T \simeq 1$ does correspond to where the vertical scale of Charney-type baroclinic instabilities first become of the same order as the





pressure scale height and altitude of the tropopause (e.g. Branscombe, 1983). This would suggest that the baroclinic component of the eddies need to span a large fraction of the active troposphere in order to be able to exert a strong influence on its
thermal structure. But it does, therefore, indicate that when this condition is satisfied, it can exhibit a similar form of baroclinic adjustment to that found in the laboratory.

It would clearly be of interest to explore this trait in more detail in other, more realistic, GCM studies that also include the effects of radiative transfer and latent heat transport.

## 4 Baroclinic and barotropic instabilities in the Solar System

As discussed above in Section 1, the Earth's atmosphere satisfies the CSP necessary conditions for instability mainly through criterion *(iii)*, associated with persistent equatorward temperature gradients at the surface and the planetary vorticity gradient in the free atmosphere. The flanks of the upper level mid-latitude jet might also lead to local changes of sign of $\partial Q/\partial y$ in the interior, while horizontal thermal gradients at the tropopause might also allow the CSP conditions to be satisfied at times through criterion *(iv)* (e.g. see Vallis, 2017, section 9.9). These factors suggest multiple ways in which large-scale instabilities
might occur in the Earth's atmosphere, so which mechanism (baroclinic or barotropic) is likely to dominate?

Sections 2.2 and 3 both highlight the importance of the dimensionless parameters, $\mathcal{B}u$ and $\mathcal{R}o_T$, in favouring which mechanism is likely to be most important, although the distribution of $\partial Q/\partial y$ and its sign changes in the horizontal or vertical also plays a role. Table 2 presents some rough estimates of the values of $\mathcal{B}u$ and $\mathcal{R}o_T$ for the Earth, Mars, Venus and Jupiter, based on Eqs (12) and (14). These parameters for Earth are typically much less than O(1), that would tend to suggest that baroclinic
instabilities dominate, as is well known (e.g. James, 1994; Vallis, 2017).

**Table 2.** Estimates of the main dimensionless parameters for the Earth, Mars, Venus and Jupiter. Radiative time constant estimates are based on values from NASA's PDS (See https://pds-atmospheres.nmsu.edu/education_and_outreach/encyclopedia/radiative_time_constant.htm).

| Planet | $\mathcal{B}u$ | $\mathcal{R}o_T$ | $\tau_R$ (s) | $\mathcal{T}_R$ |
|---|---|---|---|---|
| Earth | 0.02 | 0.06 | $4.1 \times 10^6$ | $3 \times 10^{10}$ |
| Mars | 0.04 | 0.20 | $1.9 \times 10^5$ | $1.3 \times 10^5$ |
| Venus (planet) | 140 | 370 | $2.2 \times 10^5$ - $4 \times 10^9$ | $7.5 \times 10^{-5}$ - $8 \times 10^{12}$ |
| Venus (cloud tops) | 0.04 | 0.10 | $2.2 \times 10^5$ | $10^3$ |
| Jupiter (cloud tops) | $8 \times 10^{-4}$ | $10^{-3}$ | $1.4 \times 10^8$ | $1.5 \times 10^{18}$ |

This is confirmed in typical calculations of the Lorenz energy budget for the Earth's atmosphere. Figure 12(a) shows the result of a typical Lorenz energy cycle, as computed by Boer and Lambert (2008), with energy reservoirs shown in units of J m$^{-2}$ and conversion rates in W m$^{-2}$. This clearly shows that much of the convertible energy in the atmosphere resides in the AZ reservoir, which hosts more than twice as much as the rest of the reservoirs put together. The strongest internal conversions
are the CA and CE terms, which indicate a net transfer of energy from zonal potential energy AZ to KE via AE, which is the





classical route representative of baroclinic instability (cf Fig. 1(d)). The kinetic energy conversion, CK, is relatively small but robustly negative, indicating a transfer from KE to KZ consistent with the driving of a zonal mean eddy-driven jet stream at mid-latitudes (e.g. James, 1994; Vallis, 2017). The AZ reservoir is maintained through the GZ term, representing the effects of differential radiative heating and cooling, while energy is ultimately removed from the system via the dissipative flux terms FZ

and FE.

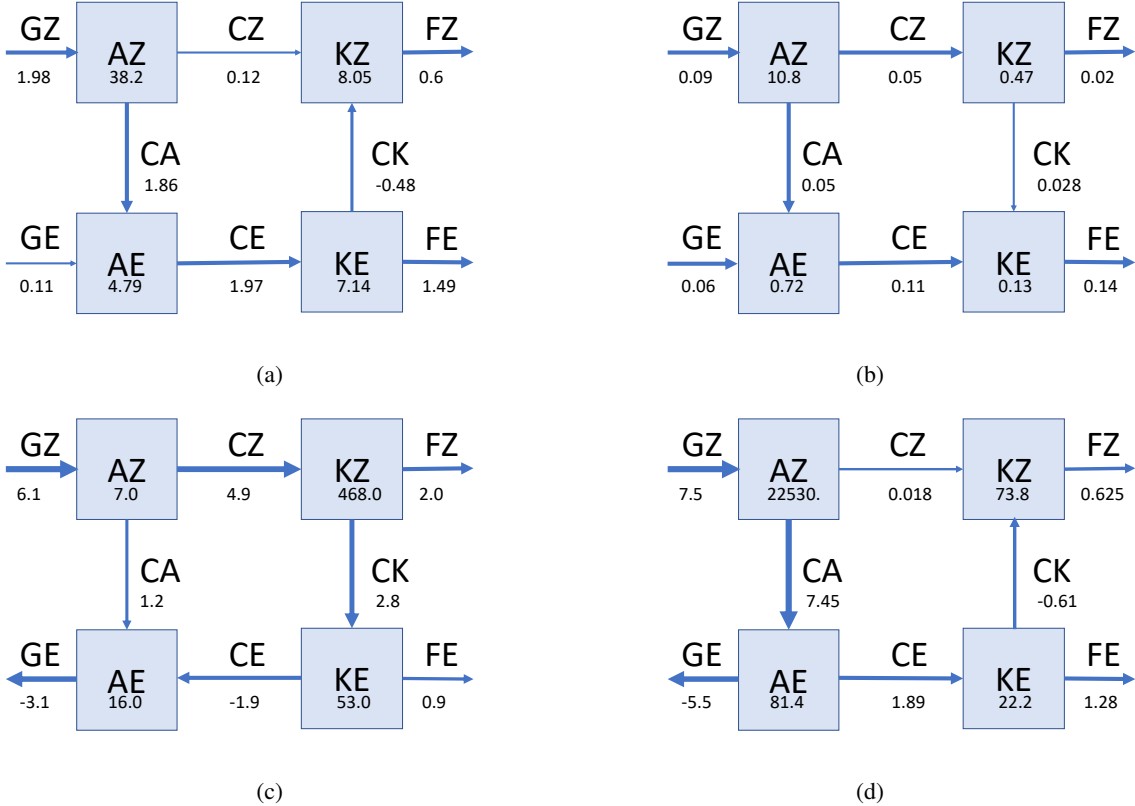

**Figure 12.** Representations of energy storage and interconversions in the form suggested by Lorenz (1955) and Lorenz (1967) for a selection of the main planets of the Solar System: (a) Earth (calculations from Boer and Lambert, 2008); (b) Mars (calculations from Tabataba-Vakili et al., 2015); (c) Venus (calculations from Lee and Richardson, 2010) and (d) Jupiter's weather layer (from preliminary calculations using a higher-resolution version of the model described in Young et al., 2019a). As before, energy reservoirs are shown in units of $10^5$ J m$^{-2}$ and conversion rates and source/sink terms in units of W m$^{-2}$.

This pattern of energy reservoirs and fluxes is well known for the Earth, and is broadly consistent with the corresponding Earth-like case discussed in Section 3. But what about other planets in the Solar System with substantial atmospheres which may be in very different circulation regimes from Earth? Where might they fit in comparison to the scenarios indicated by the regimes found with simplified GCMs, such as in Section 3?





### 4.1 Mars

Mars is arguably the most Earth-like planet elsewhere in the Solar System, at least so far as its atmosphere and climate are concerned. It is roughly half the linear size of Earth (radius $a \simeq 3400$ km) with a rotation period of just over 24 hours. It lies at a distance of around 1.3-1.5 Astronomical Units (AU) with an orbital period of around 687 days and, with a rotation axis tilted by approximately $25°$ from the perpendicular to its orbit, experiences a strong seasonal cycle much like the Earth. It possesses an atmosphere mainly consisting of $CO_2$ with a surface pressure of around 6 hPa which, although much less than on Earth, is sufficient to interact strongly with the rocky surface. Both water and $CO_2$ can condense as ices to form clouds and surface deposits.

Its temperature structure and rapid planetary rotation lead to relatively small values of $\mathcal{R}o_T$ and $\mathcal{B}u$ (see Table 2), though not as small as for the Earth. Its relatively short radiative time constant leads to a value of $\mathcal{T}_R$ which is much larger than O(1) but not hugely so, indicating relatively weak radiative damping compared with Coriolis forces.

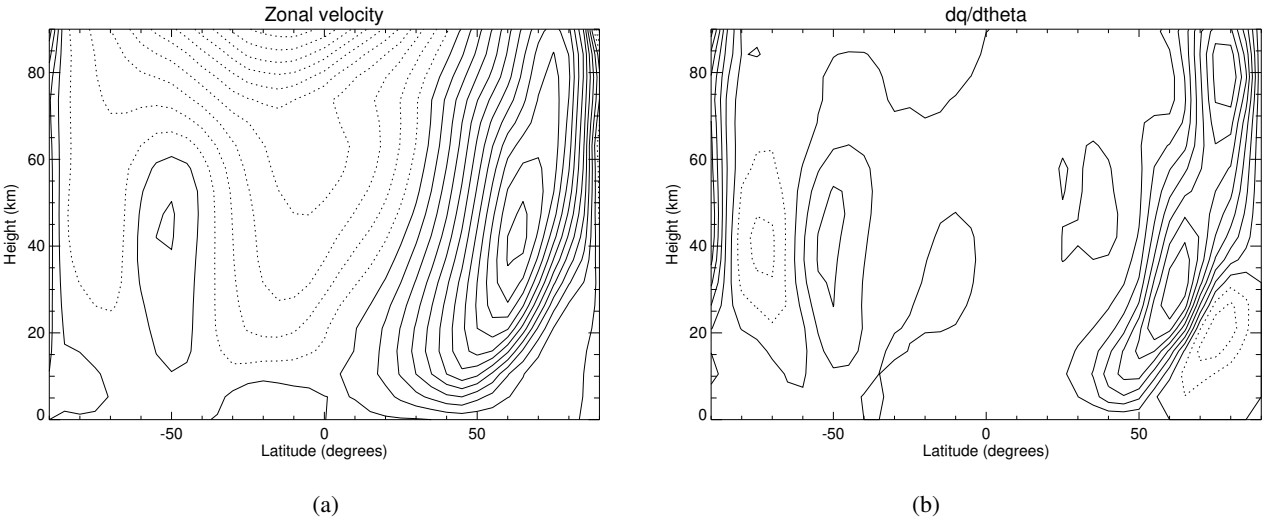

(a)                                        (b)

**Figure 13.** Zonal mean sections of (a) zonal wind (m s$^{-1}$) and (b) $\partial Q/\partial \phi$ (s$^{-1}$ rad$^{-1}$) typical of Martian northern hemisphere winter. Data were taken from the Mars Climate Database (e.g. Lewis et al., 1999, see also http://www-mars.lmd.jussieu.fr/mars/access.html) for $L_s = 270°$ using the standard Mars climatology. Contour intervals are (a) 15 m s$^{-1}$ and (b) 0.0002 s$^{-1}$ rad$^{-1}$ and negative contours are shown dashed.

#### 4.1.1 PV structure

Its similarities with Earth are also reflected in its potential vorticity structure, as can be seen e.g. in Figure 13, which shows the zonal mean zonal wind and QGPV gradient for a typical atmospheric state during northern hemisphere winter on Mars, from data obtained from the Mars Climate Database (e.g. Lewis et al., 1999, ; see also http://www-mars.lmd.jussieu.fr). As can be seen in Fig. 13(a), zonal winds are typically concentrated into deep, mid-latitude jet streams during winter and the





adjacent seasons, with strong vertical and horizontal shears and a pronounced tilt with latitude. The corresponding structure in QGPV (see Fig. 13(b)) shows $\partial Q/\partial y$ ($\partial Q/\partial \phi$) positive in the jet itself (cf Fig. 13(a)) but with weaker reversals in sign on either side. Together with the positive equatorward temperature gradient at the surface in the north, this would indicate that the CSP conditions can be satisfied in this season either by criterion *(i)* or *(iii)*, much as on Earth. The atmosphere might then be

expected to exhibit either baroclinic or barotropic instabilities, or a mixture of both, although the linear instability analysis of Barnes (1984) suggests that the baroclinic instability mechanism dominates for Martian conditions while the barotropic shear mechanism acts to damp the baroclinic instability.

It is interesting to note that, at this time during southern summer, the corresponding zonal jet stream becomes much weaker and even reverses direction in places. The QGPV gradient does exhibit a weak reversal in sign, though the equatorward thermal

gradient near the surface also reverses in sign during summer, so that conditions are much less favourable for baroclinic instability via CSP criterion *(iii)*. This is consistent with the observed near suppression of baroclinic instability in Martian summers at mid-latitudes, except for occasional very shallow disturbances that are sometimes observed on small horizontal scales close to the polar cap edge (e.g. Gierasch et al., 1979b).

### 4.1.2 Energy budget

The corresponding Lorenz energy budget is needed to establish which is the dominant mechanism to energise the Martian eddy circulation. This has been computed recently by Tabataba-Vakili et al. (2015), based on an assimilated analysis of orbiting spacecraft measurements of Mars over a period of at least 3 Mars years (Montabone et al., 2014). The data were assimilated into the UK version of the LMD Mars GCM to produce a daily record of Martian meteorology over the entire period, from which the energy budget could be computed.

The results show a qualitatively similar pattern to that of the Earth, with the majority of dynamically active energy in the atmosphere stored as zonal potential energy AZ. The dominant conversion terms are the CA and CZ conversions from AZ to AE and KZ respectively (at 50 mW m$^{-2}$), and especially the CE term (at 110 mW m$^{-2}$). This indicates a strong energetic role for the thermally direct axisymmetric (Hadley cell) overturning circulation as well as the principal baroclinic conversions CA and CE maintaining synoptic eddies. This confirms the role for baroclinic instability as suggested from the configuration of

potential vorticity gradients in Fig. 13(b). The barotropic eddy-zonal conversion CK, however, is positive and around half the amplitude of CA, suggesting a role for exchanges consistent with a mixed baroclinic-barotropic instability. The GE source term for AE is also quite substantial and of the same order as CA and CZ. This is because the day-night contrast in solar heating on Mars generates a strong thermal tide response which contributes to the AE and KE reservoirs, although this varies significantly with season and the amount of dust loading in the atmosphere. As discussed by Tabataba-Vakili et al. (2015), this budget does

exhibit fairly significant seasonal variations, also between the northern and southern hemispheres, but the overall structure of the energy budget remains largely unchanged.



## 4.2 Venus

Venus is the other main terrestrial planet in the inner Solar System that possesses a substantial atmosphere. It is also Earth-like in some respects, but very different in others. Around the same size as Earth, Venus is somewhat closer to the Sun than Earth

at a mean distance of 0.72 AU and orbits the Sun in around 224.7 days. The planetary rotation period, however, is very long compared to Earth, with a sidereal rotation period of 243.7 days in a retrograde sense and a very small obliquity angle (177.3°, equivalent to 2.6° if it rotated in the same sense as its orbit).

Its atmosphere is deep and massive, consisting mainly of $CO_2$ with a surface pressure of more than 90 bars and typical surface temperature of 740 K. The entire planet, however, is almost completely shrouded in thick clouds, thought to consist

mainly of aqueous sulphuric acid droplets that scatter sunlight very efficiently (with a Bond albedo of 0.77) and located between 40 and 60 km above the surface. Dynamically, Venus is in a very different regime to the Earth. Its winds are dominated by a very rapid super-rotation with maximum zonal winds of more than 100 m s$^{-1}$, located around or slightly above the main cloud decks at altitudes of 50-70 km. This means that the atmosphere at the cloud level rotates around 60 times faster than the underlying surface. The processes driving such a remarkable circulation are still imperfectly understood, though likely entail

the role of waves and eddies on various scales (e.g. see Read et al., 2018; Sánchez-Lavega et al., 2017, for recent reviews). Despite such strong winds and dynamical activity, horizontal temperature gradients are relatively weak compared with those in the vertical, with typical equator to pole thermal contrasts of no more than 10-20K.

These factors lead to somewhat different estimates of some of the key dynamical parameters, depending upon whether the atmosphere is viewed in the frame of the underlying planet or in corotation with the main cloud deck. In the frame of the planet,

values of $\mathcal{B}u$ and $\mathcal{R}o_T$ are much larger than O(1), indicating that the flow is unlikely to be geostrophic (its cloud level zonal winds are predominantly in cyclostrophic balance with centrifugal accelerations dominating over Coriolis accelerations; e.g. see Sánchez-Lavega et al. (2017)) or baroclinically unstable, although it could be consistent with various forms of barotropic shear instability. If the circulation is viewed in the average frame of the cloud level winds (e.g. see Young et al., 1984), however, which rotate around the planet in around 4 days, a different perspective emerges with typical values of $\mathcal{B}u$ and $\mathcal{R}o_T$ of around

0.04-0.1. In this frame, therefore, the cloud level circulation may appear to be in quasi-geostrophic balance with the possibility of baroclinic instabilities that would likely be localised in the vertical around the levels of the main cloud decks, provided the potential vorticity configuration would allow the CSP necessary conditions for instability to be satisfied.

### 4.2.1 PV structure

A key difficulty in determining the PV structure of Venus's atmosphere arises from a lack of detailed observations of winds

and temperature within and beneath its main cloud decks (e.g. see Sánchez-Lavega et al., 2017). It is necessary, therefore, to fall back on numerical models of Venus's atmospheric circulation for the kind of information needed to compute quantities such as $\partial Q/\partial y$. But the numerical simulation of Venus's atmospheric circulation is still at a relatively immature state, with realistically forced models appearing only recently (e.g. Lebonnois et al., 2010, 2016; Mendonça and Read, 2016). Simpler





models have been studied more extensively but have struggled to reproduce key features of the circulation and have exhibited

significant divergence among different model formulations (e.g. Lebonnois et al., 2013; Lewis et al., 2015).

For the present purpose, therefore, we follow Young et al. (1984) and Sugimoto et al. (2014) in examining a somewhat idealised form of the zonal mean state of a Venus-like atmosphere. This is illustrated in Figure 14, taken from the work of Sugimoto et al. (2014), and shows a zonal mean section of the zonal wind (Fig. 14(a)) and corresponding field of $\partial Q / \partial \phi$ (in the average frame of the zonal wind at an altitude of 56 km; Fig. 14(b)). The zonal wind section (in the planetary reference

frame) captures the main features of the observed zonal winds on Venus, with a monotonic growth of zonal wind strength with altitude at all latitudes towards maximum values of 100-120 m s$^{-1}$ at around 70 km altitude, but with the formation of jet-like features at mid-latitudes in both hemispheres. This idealisation then neglects any vertical shear above the cloud tops, although this may not be particularly realistic. However, the flow structure around the cloud top altitudes are of most interest here. The winds are constructed to be in gradient wind balance (between horizontal pressure gradients and a combination of Coriolis and

centrifugal "forces") with the corresponding pressure and temperature fields.

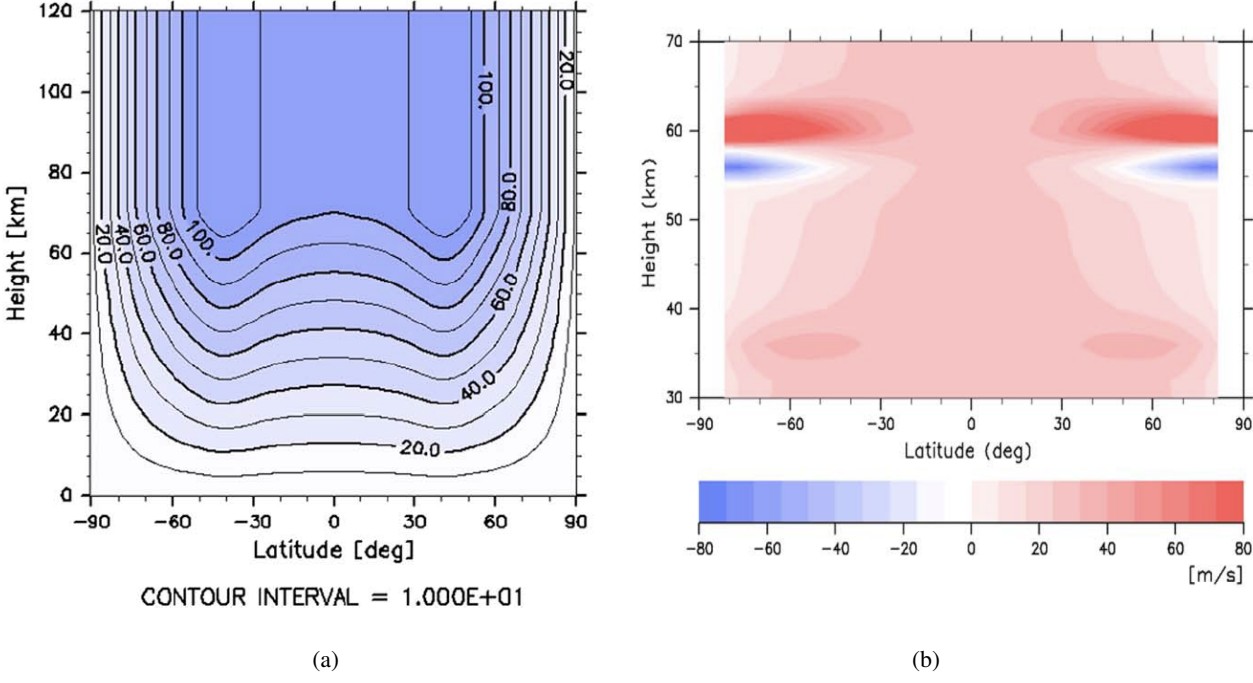

(a)                                              (b)

**Figure 14.** Zonal mean sections of (a) zonal wind (m s$^{-1}$) and (b) $a\partial Q / \partial \phi$ (m s$^{-1}$) typical of Venus's global circulation, as presented by Sugimoto et al. (2014) from their numerical model simulations.

The distribution of $\partial Q / \partial \phi$ in Fig. 14(b) indicates a concentration of PV gradient at high latitudes, especially within the cloudy layers between altitudes of 55 and 65 km, (although the clouds are not represented explicitly in this model). Moreover, this gradient clearly changes sign in the vertical while remaining positive over most of the domain at lower levels. This would





immediately suggest the possibility of a baroclinic instability localised within the cloud deck levels by satisfying the CSP

conditions for instability via criterion *(i)*. A change of sign in the horizontal is only discernible in the layer around 56 km, however, suggesting that barotropic instabilities might also be possible, though again centred mainly within the cloud decks. Nearer the surface, $\partial Q/\partial \phi > 0$ but surface temperature gradients are very weak, suggesting that Venus may be able to satisfy the CSP condition via criterion *(iv)* only marginally. Besides, the flow is far from being in geostrophic balance at these levels, although that would not necessarily rule out the possibility of a non-geostrophic form of baroclinic instability.

**4.2.2 Energy budget**

The energy budget for Venus is similarly not available from observations and so we must rely on model simulations to compute the likely energies and exchanges. To the authors' knowledge, this has not so far been done for any realistically forced, comprehensive GCM of Venus that faithfully reproduces the observed winds and circulation. However, Lee and Richardson (2010) did compute Lorenz energy budgets for a set of somewhat simplified GCM simulations of Venus-like circulations and so this

is what is presented here. However, these model simulations did not fully represent the radiative forcing of the atmosphere, including its diurnal cycle, nor did they fully capture the observed distribution of angular momentum in the simulations. So the budget shown here is likely to be inaccurate in some degree and missing some key processes. Nevertheless, the results are worthy of attention as a first step towards a more accurate calculation.

Figure 12(c) presents the energy budget calculations for the spectral core case of Lee and Richardson (2010) with full

damping, as being reasonably representative of this class of Venus model (without an explicit diurnal cycle). The energy reservoirs show a very different distribution of energy from either Earth or Mars, with most of the dynamically active energy in the atmosphere residing in the zonal and eddy kinetic energies, KZ and KE. The potential energy reservoirs, however, are certainly not negligible and suggest a significant concentration of available potential energy in the AE reservoir. AZ is relatively small, though, as would be expected for an atmosphere with relatively weak meridional temperature gradients. Among the

conversion terms, CZ is the largest, indicative of a strong, thermally-direct Hadley overturning circulation to maintain the very large KZ reservoir. The next largest conversion is CK, which evidently acts in the positive direction to convert from KZ to KE. This would suggest a dominance of barotropic processes, much as found in the idealised Earth-like simulations of Section 3 at low values of $\Omega^*$.

CA, on the other hand, also acts in a positive direction (from AZ to AE), indicating some alignment with baroclinically

active processes and perhaps suggesting a mixed baroclinic/barotropic character to the dominant eddy processes. CE, however, acts in a negative direction, from KE to AE, which would suggest that mechanical exchanges between eddy potential and kinetic energies dominate over buoyancy driven instabilities. The one term that may be anomalous when compared with Venus is the potential energy source/sink GE. This appears to be acting to remove energy from AE at a fairly strong rate. But this model takes. no account of the diurnal cycle, and hence the input of thermal energy to the thermal tides. More realistic model

simulations suggest that the thermal tides in and above the main cloud decks play an important role in maintaining the observed very strong atmospheric super-rotation (e.g. Sánchez-Lavega et al., 2017). It seems more likely, therefore, that a more complete





simulation of the Venus circulation would include a large positive contribution to GE, much as indicated for Mars in Fig. 12(b). This is a question that should be followed up urgently through analysis of more realistic model simulations.

### 4.3  Jupiter

Jupiter is the largest of a quite different class of planet within the Solar System, known as the gas giants. As this name would suggest, these planets are much larger than the Earth (Jupiter's mass is around 316 times larger than Earth's) and largely composed of hydrogen and helium in proportions similar to the Sun. They therefore do not support a solid surface, unlike the terrestrial planets, but remain fluid to great depths until reaching a core of heavier elements close to the planetary centre. At a certain depth, however, the highly compressed fluid becomes electrically conducting and at even greater depths the hydrogen

undergoes a phase change from a supercritical molecular fluid to a liquid metal (at least for Jupiter and Saturn, though not for the so-called ice giants, Uranus and Neptune; e.g. see Guillot (2005)). All four gas and ice giant planets orbit far from the Sun, with Jupiter (the innermost of these planets) in an orbit of mean radius 5.2 AU and orbital period of 11.86 years.

All four gas and ice giant planets are fully cloud covered, with clouds composed mainly of chemically reduced compounds such as ammonia ice, $NH_4SH$, $H_2S$ and water (and $CH_4$ on Uranus and Neptune). Motions of these clouds reveal a very

different circulation at the tops of their tropospheres from what is seen on Earth or the other terrestrial planets. Winds are predominantly zonal and arranged in systems of alternating eastward and westward jet streams, on scales smaller than the planetary radius. Cloud bands partly align with these zonal flows but are perturbed by waves and oval vortical eddies across a range of scales. Near the main cloud decks, incoming sunlight contributes significantly to the radiative energy budget. However, both Jupiter and Saturn (and Neptune) are significant net emitters of excess thermal radiation from a heat source in the deep

interior, most likely representing residual thermal energy still escaping to space from the time of the original formation of these planets (e.g. Guillot, 2005). Both sunlight and the release of latent heat energy from the condensation of trace amounts of water in these planets play an important role in an atmospheric layer around 200-300 km thick, commonly known as the "weather layer". The zonally dominated circulation in this weather layer is not thought to penetrate very deeply below the bottom of this layer, though recent measurements from the Juno mission suggest that the zonal wind pattern could penetrate as much as 3000

km below the visible clouds (around 4% of Jupiter's radius; Kaspi et al. (2018)) though this is still somewhat uncertain (e.g. see Kong et al. (2018)).

Despite their massive sizes, both Jupiter and Saturn are very rapid rotators with rotation periods around 10 hours. This would imply that differential motions within their fluid envelopes are highly likely to be strongly geostrophic. The observable regions of their atmospheres, above, within and somewhat below the main $NH_3$ and $H_2O$ cloud decks, are found to be stably stratified,

indicating the possibility of instability processes that have dynamical properties in common with those found in the Earth's atmosphere and oceans. Their rapid rotation leads to values of the main Rossby radius of deformation, $L_D$, to be much smaller than the planetary radius $a$. This is reflected in the very small values of $\mathcal{B}u$ and $\mathcal{R}o_T$ indicated for Jupiter's cloud tops in Table 2. Together with the very large value of $\mathcal{T}_R$, this would seem to suggest that geostrophic forms of barotropic, and possibly baroclinic, instabilities are likely to occur in the weather layers of these planets.





### 4.3.1 PV structures: observations

The occurrence of such instabilities, however, depends crucially on the distribution of PV within the weather layers of these planets. As with the other planets under discussion here, Jupiter's atmosphere needs to satisfy the CSP necessary condition for instability somehow. Without a solid surface, however, CSP criteria (iii) cannot be satisfied, unlike on Earth or Mars. Such an argument was used by Gierasch et al. (1979a) to suggest only a minor role at most for baroclinic instabilities on Jupiter. But as subsequently noted by Conrath et al. (1981), the possibilities of satisfying the CSP condition through criteria (i) or (ii) remain, especially if Jupiter or Saturn possess a strong tropopause which can act as an internal interface that supports a horizontal thermal gradient. The analysis of Conrath et al. (1981) noted that, in the absence of significant horizontal shear and with a lower boundary that was essentially isentropic and weakly stratified, westward jets could satisfy the CSP condition through criterion *(i)* (or *(ii)* if the tropopause is considered as an upper boundary or interface). Their calculations indicated that such an "inverted Charney" form of baroclinic instability could have relatively fast growth rates and even lead to up-gradient momentum fluxes for instabilities on a small enough horizontal scale.

The vorticity structure of the zonal mean flow at the cloud tops has been measured since the time of the Voyager spacecraft encounters with Jupiter in 1979. By tracking features in the ubiquitous ammonia cloud decks in successive images taken by the spacecraft, the pattern of eastward and westward jet streams could be derived as a profile in latitude from which the vorticity and its northward gradient could be obtained (e.g. Ingersoll et al., 1981; Salyk et al., 2006; Galperin et al., 2014). These profiles indicate that the northward gradient of *absolute* vorticity robustly changes sign with latitude in a number of places, typically around the peaks of westward jets, leading to some suggestions that these jets may be prone to barotropic instabilities since they would appear to satisfy criterion *(i)* for the CSP instability condition (Ingersoll et al., 1981). The stretching term associated with the thermal structure of the flow may also be important, leading to some suggestions (e.g. Ingersoll and Cuzzi, 1969; Scott and Dunkerton, 2017) that vigorous lateral mixing of PV could result in a monotonic, staircase-like variation of $Q$ with latitude which would be unconditionally stable to shear instabilities. Such a process that might hold the atmosphere close to a state of marginal instability would effectively constitute a form of *barotropic adjustment* by analogy with Stone's baroclinic adjustment hypothesis (Stone, 1978).

Observational studies that have combined wind measurements from cloud motions with remotely sensed temperature retrievals (Read et al., 2006, 2009a; Antuñano et al., 2019) and a geostrophic balance assumption for both Jupiter and Saturn, however, also indicate significant reversals of $\partial Q/\partial y$. Figure 15(a) shows an example of a section of $\partial Q/\partial y$ in latitude and height in the southern hemisphere of Jupiter, based on data from the Cassini spacecraft (Read et al., 2006). This clearly shows patches where $\partial Q/\partial y < 0$ overlying westward jets but persisting over significant ranges in altitude. $\partial Q/\partial y$ even appears to change sign in the vertical direction in some locations in Jupiter's stratosphere, although these observations should be treated with caution since the reconstruction of the flow at high altitudes requires the integration of the geostrophic thermal wind shear relation which can amplify the effects of measurement noise and uncertainties, especially in quantities that require spatial differentiation (e.g. see Read et al., 2009a). Being based on nadir observations, the retrieved thermal measurements also have


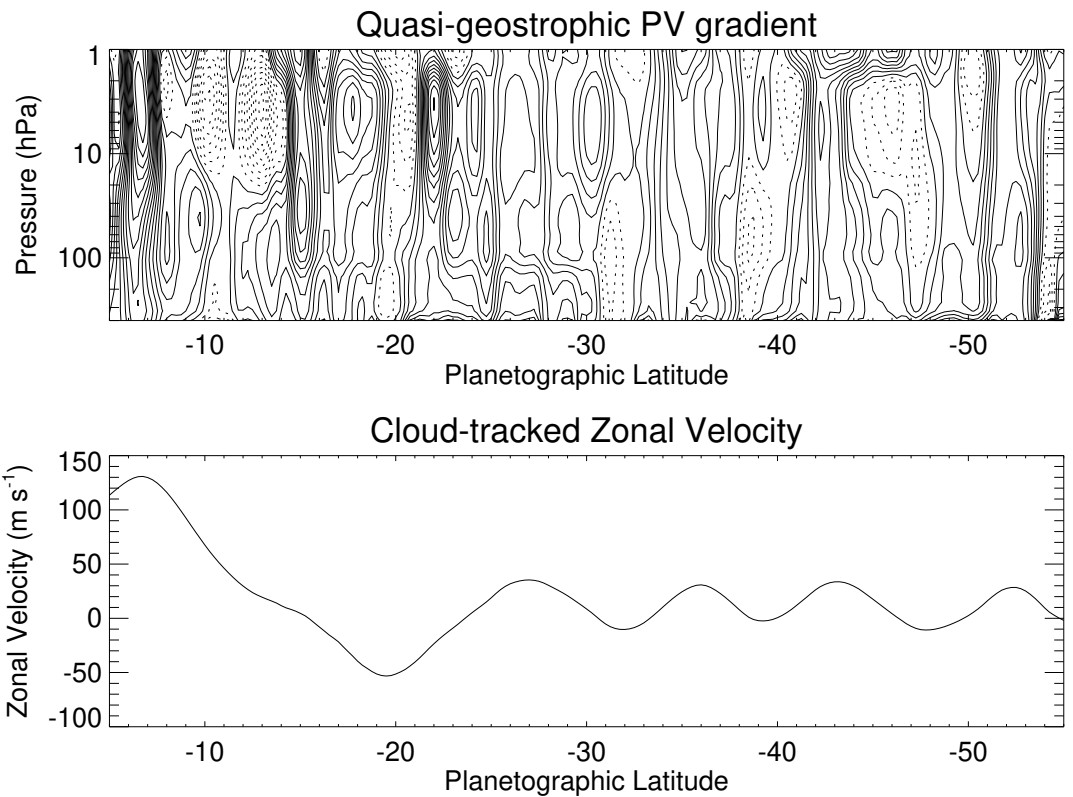

**Figure 15.** (a) Zonal mean section of $\partial Q/\partial y$ (s$^{-1}$ m$^{-1}$) and (b) zonal mean zonal wind (m s$^{-1}$) in the southern hemisphere of Jupiter, derived from Cassini observations by Read et al. (2006).

rather limited vertical resolution. So it cannot be ruled out that structures associated with sharp thermal gradients along the lines suggested by Scott and Dunkerton (2017) might exist that would eliminate such reversals in sign of $\partial Q/\partial y$.

The distribution of $Q$ beneath Jupiter's cloud tops, however, is virtually unknown because of a dearth of detailed measurements of winds, temperatures or composition. But there seems no reason *a priori* to conclude that internal changes of sign of $\partial Q/\partial y$ in the deep troposphere, or of $\partial T/\partial y$ at the tropopause, cannot exist, and that therefore the CSP necessary condition for baroclinic or barotropic instability could be satisfied through criteria *(i)* or *(ii)*.

### 4.3.2 PV structures: GCMs

A number of studies have appeared recently in which fully three-dimensional, time-dependent GCMs have been developed of Jupiter's or Saturn's weather layers which seek to capture the thermal and vorticity structures of the flow around and beneath the cloud tops (e.g. Liu and Schneider, 2010; Lian and Showman, 2008, 2010; Palotai et al., 2014; Young et al., 2019a, b; Spiga et al., 2020). These models have included the effects of solar heating in the stratosphere and upper troposphere and upwelling


internal heating from the deep interior, and have been capable of capturing a number of realistic features such as the equatorial prograde jets and multiple eddy-driven zonal jets at mid-latitudes.

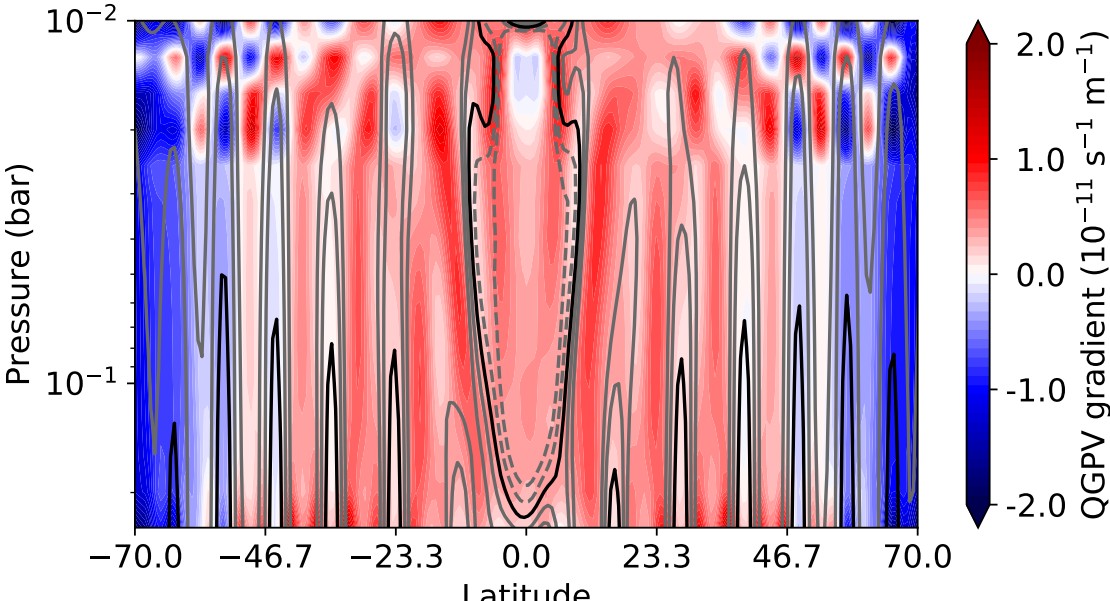

**Figure 16.** Zonal mean latitude-height section of $\partial Q/\partial y$ (s$^{-1}$ m$^{-1}$) and zonal mean zonal wind (m s$^{-1}$) in a numerical simulation of Jupiter's weather layer, from Young et al. (2019a). Data correspond to run B of Young et al. (2019a) and were averaged over that last 100 days of the equilibrated run. Shading represents $\partial Q/\partial y$ and the line contours represent zonal mean zonal velocity $\overline{u}$, shown for -10, -5, +5 and +10 m s$^{-1}$ in grey and 0 m s$^{-1}$ in black.

Figure 16 shows a cross-section of $\partial Q/\partial y$ and $\overline{u}$ from a GCM simulation of Jupiter's weather layer by Young et al. (2019a) (their run B, with a horizontal resolution of $512 \times 256$ points in longitude and latitude). As with the observations, this shows a generally clear correlation between $\overline{u}$ and $\partial Q/\partial y$ in the upper troposphere but with some sign reversals of $\partial Q/\partial y$ in both the horizontal and vertical in relation to both westward *and eastward* jets in several cases. This would seem to confirm that the

simulated flow sustains the necessary conditions for baroclinic instability through CSP criterion *(i)* as discussed above.

### 4.3.3 Energy Budget

A preliminary Lorenz energy budget computed using an updated, higher-resolution version of the Jupiter weather layer model described by Young et al. (2019a) is shown in Fig. 12(d). This represents the model's attempt to capture the energetics of a Jupiter-like weather layer between the pressure levels of 18 bars to 10 hPa. The energy reservoirs reflect a distribution of

potential and kinetic energy typical of a rapidly rotating, stratified atmosphere, with most of the stored energy residing in the





zonal mean potential energy AZ. Smaller amounts are found in the KZ and AE reservoirs with KE the weakest, though still containing much more energy per unit area than the corresponding reservoirs for Earth.

Energy conversions, however, are remarkably Earth-like in their relative sizes and directions, with CA and CE as the largest conversion rates in the sense expected for baroclinic instabilities. This confirms the indication from the PV structure that anticipated that baroclinic instabilities could occur and play a significant role in the circulation. CK acts to maintain the zonal mean flow through transfers from KE, at a vertically integrated rate (0.61 W m$^{-2}$) that is not unreasonable with regard to the observed transfer rate near the cloud tops (e.g. Ingersoll et al., 1981; Salyk et al., 2006; Galperin et al., 2014). CZ acts in the thermally direct sense but is relatively weak, reflecting the comparatively minor role played by zonally symmetric overturning circulations in the global circulation. The largest terms overall are CA, GZ and GE, indicating some very strong forcing of the zonal mean potential energy field and conversion to eddy potential energy, much of which is then extracted by diabatic processes. A complication is that this model simulation includes both latent heat fluxes and exchanges associated with a moist convection parameterization, which may not be fully accounted for in the conventional Lorenz energy budget approach. But the overall structure of this cycle strongly suggests an important role for baroclinic instability, at least in this model simulation of Jupiter's weather layer.

# 5 Conclusions

Throughout this review it has been clear that baroclinic and/or barotropic instabilities play a key role in generating some of the most energetic eddies and waves in most rotating, stratified systems, including systems that are very different from the Earth's atmosphere and oceans. The distinction between baroclinic and barotropic instabilities is, to a large extent, an artificial one, based largely upon whether the dominant conversion of energy from an initial basic state is from its stored potential or kinetic energy. But the CSP necessary condition for instability embraces both mechanisms, which are governed by similar vorticity constraints. In practice, eddy formation in realistic systems with both vertical and horizontal shear could entail contributions from both potential and kinetic energy stored in the basic state, so that it is natural to consider either mechanism as different facets of a common process. This suggests the value of a unifying approach, although a number of outstanding issues remain.

## 5.1 Stability theory

The body of stability theory continues to provide important and useful guidance to help predict the onset of instabilities in various practical geophysical and astrophysical contexts, and in particular to establish conditions for when instabilities will not occur. But there are still many situations that are not fully worked out in complete generality and most published stability criteria have shortcomings that fail to take certain factors into account. After more than 50 years, the CSP instability condition is still widely used to assess the stability of zonal flows in atmospheres and oceans, even though it is based on linearised quasi-geostrophic theory for perturbations to a purely axisymmetric basic state. As an instability condition it only represents a necessary condition but not a sufficient one to guarantee instability if it is met.




The more sophisticated stability theorems of Arnol'd (1966) and more recent extensions (e.g. see Mu and Wu, 2001, for a review) offer a more refined approach which takes some account of nonlinearity and non-zonal structure, although with some non-trivial restrictions. It is most well developed for quasi-geostrophic flows but a full treatment e.g. of the primitive equations

of dynamical meteorology is still lacking. Much of the most well developed theory also applies strictly to a Hamiltonian system, neglecting forcing and dissipation processes that may maintain the flow in a quasi-equilibrium state. Some exceptions exist (e.g. Andrews, 1984a, b) but they are still quite rare.

## 5.2 Equilibration and adjustment

Linear stability theory generally applies to the behaviour of small amplitude perturbations, but real flows are more typically

observed when any incipient instabilities have grown to large amplitude and saturated. Precisely how equilibration is achieved may depend strongly on the structure of the flow and how it is maintained and/or dissipated. Even weakly nonlinear theories (e.g. Pedlosky, 1970, 1971; Drazin, 1970) show that growing instabilities act to reduce the vigour of the instability and its growth rate by modifying the structure of the initial background flow to make it less strongly unstable. For barotropically or baroclinically unstable flows, this commonly entails a combination of reducing the strength of any gradients of PV or buoyancy

in the base state via a pattern of eddy fluxes, but the structure of the end state may depend on a variety of factors and constraints, including the strength and distribution of any forcing.

Where the forcing and dissipation is relatively weak, such that the characteristic timescale $\tau_H$ for the system to respond to changes in forcing is much longer than any advective overturning timescale $\tau_A \sim L/U$, fluid motions may be able effectively to nearly neutralise the instability so that, on timescales $O(\tau_H)$ or longer, the flow stays close to a marginally unstable state. This

scenario may offer the possibility of parameterizing the effects of strong barotropic or baroclinic instabilities on large-scale flows through forms of barotropic or baroclinic adjustment (by analogy with convective adjustment). However, simply posing a requirement to seek a marginally unstable state is usually insufficient to determine or predict from first principles a unique equilibrated state unless additional constraints or information are available, such as an applicable optimisation principle (e.g. for entropy production, potential enstrophy or a generalised free energy, e.g. Lucarini (2009), Hollerbach et al. (2004), Bouchet

and Venaille (2012)).

Such an adjustment principle may have useful applicability to the interpretation of observed flow systems, however. Dowling (1993) and Stamp and Dowling (1993), for example, have suggested that the zonal mean winds in the observable cloud level atmospheres of both Jupiter and Saturn may approach a state that is just marginally unstable with respect to a form of Arnol'd's second stability theorem, a hypothesis that seemed to be confirmed in more quantitative observations of potential vorticity

(Read et al., 2006, 2009a). This was used by Read et al. (2009b) to infer the existence of a unique reference frame on each planet which corresponded closely on Jupiter to that of its magnetic field and deep interior. For Saturn, however, it differed significantly from the magnetic rotation periods inferred from measurements from the Voyager or Cassini spacecraft, but agreed more closely with alternative estimates of Saturn's interior rotation from measurements of its gravity field or perturbations in its rings (Anderson and Schubert, 2007; Mankovitch et al., 2019).



## 5.3 Baroclinic atmospheres

For baroclinic atmospheres, such as that of the Earth, the possibility that its zonal mean circulation is at least partly controlled by a form of baroclinic adjustment (Stone, 1978), although debates continue to the present day as to whether this is a valid assumption and/or precisely what principle is being optimised during equilibration (e.g. Zurita-Gotor and Lindzen, 2007; Schneider, 2007). Flaws in Stone's original argument that hinges on a tendency for the flow to deviate only slightly from a marginally unstable state were discussed above, leading to alternative suggestions that the flow evolves towards a state with reduced nonlinear interactions (e.g. Schneider and Walker, 2006). Such a hypothesis is consistent with the observation that equilibrated flows in both the laboratory and in the atmospheres of Earth and some other planets exhibit the property that potential vorticity $Q \simeq Q(\psi)$.

The tendency shown in both laboratory experiments and simple GCM simulations for the total meridional heat transport (eddy plus zonal mean) to be largely independent of rotation rate (see Sections 2.2 and 3) over the range in $\mathcal{B}u$ and/or $\mathcal{R}o_T$ from O(1) to $\sim 0.05$ is somewhat remarkable. It is apparently due to a mutual compensation between decreasing heat transport by the zonal mean flow and increasing eddy heat transport with $\Omega$ until the latter reaches a peak around $\mathcal{R}o_T \simeq 0.1$. The resulting impact on the large-scale zonal mean flow is for the thermal structure to remain fairly static over this range of $\Omega$, which would imply that the mean slope of the isentropes in the latitude-height plane at mid-latitudes also remains largely independent of $\Omega$.

This is consistent with a roughly constant value of the supercriticality parameter

$$\xi = \frac{f}{\beta} \frac{\partial \bar{\theta}/\partial y}{\Delta \theta_v} = -\frac{f}{\beta H} \frac{\partial \theta/\partial y}{\partial \theta/\partial z}, \tag{15}$$

where $\Delta \theta_v$ is a vertical contrast in potential temperature and $H$ is a vertical length scale, often taken to be the pressure scale height of an atmosphere. $\xi$ is thus a measure of the mean slope of the isentropes, which many studies have indicated remains close to O(1) over a wide range of parameters centred around those for Earth (e.g. Stone, 1978; Schneider and Walker, 2006; Zurita-Gotor and Lindzen, 2007). A number of different implications for $\xi \simeq 1$ have been noted, including an absence of a significant inverse kinetic energy cascade, since $\xi$ can also be interpreted as measuring the squared ratio of the Rhines scale

$$L_R = \left( \frac{U}{\beta} \right)^{1/2}, \tag{16}$$

(based on the zonal mean thermal wind scale $U \sim U_T$) which can estimate the large-scale limit to the main inverse cascade, to the Rossby deformation radius $L_D$ (although $L_R$ is more conventionally estimated using $U \sim (EKE)^{1/2}$ which can differ significantly from $U_T$ under some conditions). This would also suggest a suppression of nonlinear wave-wave interactions, which would be consistent with equilibration towards a state in which $\mathbf{u} \cdot \nabla Q \simeq 0$, implying that $Q \simeq Q(\psi)$ in a quasi-geostrophic flow (where $\psi$ is the horizontal stream function). Something close to this configuration of $Q$ and $\psi$ occurs within the regular baroclinic wave regime in rotating annulus experiments (e.g. Read et al., 1986), and even the Earth's atmosphere seems to approach such a state to some extent (e.g. Butchart et al., 1989).

A possible explanation for many of the recently published results on supercriticality and baroclinic adjustment, therefore, may be related to the extent to which variations in eddy *and zonal mean* heat transports mutually compensate each other as





parameters are varied so that the total heat transport is maintained at a nearly constant rate. This compensation may be most effective within the typical range $0.05 < \mathcal{R}o_T \leq 1$, within which eddy heat transports increase with decreasing $\mathcal{R}o_T$ and zonal

mean transport decreases. Beyond this limit (i.e. $\mathcal{R}o_T < 0.05$, both components of heat transport decrease as $\mathcal{R}o_T$ decreases, and the zonal mean thermal structure might be expected to relax gradually towards the radiative-convective equilibrium state. The very high rotation rate limit of $\xi$ would then depend upon the slope of the isentropes in this radiative-convective equilibrium state, but could well lie close to O(1) for Earth-like conditions. The existence or otherwise of this kind of baroclinic adjustment regime, however, depends upon the existence of a range of parameters over which eddy heat transports strengthen with $\Omega$

and certain other parameters so can compensate for changes in the zonal mean transport. While this seems to be plausible for Earth-like atmospheres, it may not be completely general and should be investigated more thoroughly in future work.

### 5.4 Planetary atmospheres

The possible existence of both baroclinic and barotropic adjustment regimes immediately raises the question as to whether any of the known planetary atmosphere circulations are actually in one of these regimes. Given the limits on $\mathcal{R}o_T$ for the

baroclinic adjustment regime as described here, the Earth itself would seem to lie close to the high $\Omega$ margins of this regime, which would be consistent with recent work highlighting its possible effects on the circulation. Mars, with a value $\mathcal{R}o_T \simeq 0.2$ (see Table 2) would also appear at first sight to be firmly within the baroclinic adjustment regime. However, it might not fulfill the other condition for the existence of a baroclinic adjustment regime, namely, that forcing timescales $\tau_S >> \tau_A$. Radiative adjustment timescales on Mars are typically only around $\tau_R \simeq 1$ day, which is comparable to $\tau_A = L/U$ based on typical

horizontal velocities and the planetary radius. Radiative forcing is therefore relatively much stronger on Mars than on Earth, so flow configurations that appear to be vigorously unstable based on criteria derived for non-dissipative flows may actually be relatively stable in practice. This was illustrated recently in the work of Seviour et al. (2017), for example, who noted that zonal mean PV at high latitudes on Mars often took the form of a persistent annular band, across which were strong reversals in the sign of $\partial Q / \partial y$. Seviour et al. (2017) were able to reproduce this behaviour in a simple model in which zonal mean forcing was

applied by relaxing to a similar PV distribution with a short relaxation timescale. Baroclinic adjustment concepts are therefore unlikely to apply to Mars in any useful sense.

The cases of slowly rotating planets, such as Venus or Titan in the Solar System, would also seem to place them outside of the limits on $\mathcal{R}o_T$ for baroclinic adjustment. Their large scale thermal structure is typically characterised by very weak horizontal thermal gradients compared to their stratification, so isentropic slopes are very small. This would indicate that $\xi << 1$ with

relatively weak eddy heat transports, at least on the planetary scale. The strong super-rotation of the atmospheres of Venus (and Titan?), however, seems to allow for a role for baroclinic instabilities in atmospheric layers that rotate fast enough to be in a quasi-geostrophic regime, with local values of $\mathcal{R}o_T \sim 0.1$ (see Table 2). Radiative relaxation timescales near the cloud tops of Venus are also relatively long compared with advective overturning timescales, so baroclinic adjustment effects might be relevant to the circumpolar regions of Venus's cloudy layers, a possibility that deserves further investigation.

Jupiter and Saturn would seem to be well outside the baroclinic adjustment regime, based simply on their values of $\mathcal{R}o_T$ and $\mathcal{B}u$ (see Table 2). Recent work has demonstrated that Jupiter at least exhibits a substantial upscale cascade of kinetic energy
at its cloud tops, both spectrally local and non-local (Young and Read, 2017). This is consistent with estimates that suggest that $L_R$ is significantly larger than $L_D$ on Jupiter (Galperin et al., 2014), indicating that Jupiter's weather layer is significantly super-critical with an effective value for $\xi$ of around 5. This would seem to be because the relevant scale for $U$ in $L_R$ is much

larger than the thermal wind scale $U_T$, with much of the kinetic energy at Jupiter's cloud tops residing in its zonal jets. Such jets may well be predominantly barotropic in structure, at least within the weather layer.

It is perhaps unsurprising, therefore, that barotropic adjustment seems to be a useful concept in the context of the gas giant planets, Jupiter and Saturn, as discussed above, with the circulation in the weather layers remaining close to a marginally unstable state, as determined by the Arnol'd II criterion. It seems reasonable that the weather layers of Uranus and Neptune

might also be in a similar regime, which could readily be investigated in both observations and models given the availability of suitable data. The quality and coverage of the original data from the Voyager encounter may not be adequate to verify these ideas by themselves, although the accuracy and latitude coverage of cloud motion tracking has improved significantly in recent years from Earth-based (ground and space) telescopes (e.g. Garcia-Melendo and Sánchez-Lavega, 2001). These issues should certainly be addressed in the context of future missions currently being planned for the ice giants.

In the context of the ever expanding discoveries of extrasolar planets, it seems highly likely that many of them will host atmospheres within which baroclinic or barotropic instabilities may play an active role. Future investigations of their possible climates and atmospheric circulations will therefore need to take fairly full account of the influence of these instabilities, for which equilibration and adjustment concepts may surely prove useful in at least some cases.

*Data availability.* Data on the Martian atmosphere can be obtained from the Mars Climate Database website http://www-mars.lmd.jussieu.

fr/mars/access.html. Data from the model simulations of Young et al. (2019a) can be obtained from Oxford University's Research Archive at https://doi.org/10.5287/bodleian:PyYbbxpk2. Data from laboratory experiments reported in Section 2 can be obtained from the authors on request.

*Author contributions.* PLR supervised much of the research reported from the University of Oxford, analysed some of the laboratory data and was the lead author of the paper. NL, FT-V, YW and RY carried out computations and analysis for model simulations and DK computed

the energy budget reported in Section 4.3. Other authors also contributed to writing the text.

*Competing interests.* We confirm that we have no competing interests arising from the work described in this article.

*Acknowledgements.* PLR, FT-V and RMBY acknowledge support from the UK Science and Technology Research Council during the course of this research under grants ST/K502236/1, ST/K00106X/1 and ST/I001948/1. This article originated from a lecture prepared by PLR for the Summer School on Waves, Instability and Turbulence in Geophysical and Astrophysical Flows in Cargèse, Corsica in July 2019.





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
