# Peer review of "BAROCLINIC AND BAROTROPIC INSTABILITIES IN PLANETARY ATMOSPHERES - ENERGETICS, EQUILIBRATION AND ADJUSTMENT"

_Nonlinear Processes in Geophysics, 2019_

## Editor Comment (EC1)

*The line numbers below are those of the version I have had access to, once printed. For some reason, the numbers seem to be shifted by one or two units from the ones on the version I visualize on my screen.*

1.  Eq. (1). An integral sign (from 0 to $H$ I presume) is missing for the integral with respect to $z$ on the left-hand side of the equation. And it might be useful to specify at this stage that $y$ is the latitudinal coordinate and $z$ the vertical coordinate.

2.  There seems to be an inconsistency as concerns the values of the thermal Rossby number $Ro_T$. The text (starting l. 388) says that the eddy meridional heat transport peaks at $Ro_T \sim 0.07$, while Fig. 11(a) shows a peak at $Ro_T \sim 0.3$ (see also Table 1, and ll. 724-730 and 748-751).

3.  Fig. 3(c). There are two curves on the figure. What is the difference between them ?

4.  Ll. 444-445, … *leads to a value of $\mathcal{T}_R$ which is much larger than O(1) but not hugely so, ….* Well, the value given in Table 2 is 1.3 x $10^5$.

5.  Ll. 710-711, … *to infer the existence of a unique reference frame on each planet* … A unique reference frame with which properties ?

6.  Ll. 461-462, … *the observed near suppression of baroclinic instability in Martian summers ….* It would be better to give appropriate reference(s).

7.  Fig. 6. What is the precise connection between the vertical coordinate (*Stability parameter*) and the thermal Rossby number ?

8.  Fig. 8. Inset. It would be preferable to say explicitly that that *Pe* refers to heat transport by the axisymmetric flow, *Pxs* to transport by the eddies, and *m* to the number of longitudinal waves.

9.  L. 100, … *for comparison, **in Section 4,** with the known properties* …

10. Eq. (15) Inconsistency of notation. $\theta$ or $\theta$ with overbar ?

11. Ll. 418-419, … *the values of $\mathcal{B}u$, $Ro_T$ **and** $\mathcal{T}_R$ […], based on Eqs **(12-14)***

12. Ll. 780-781. Contrary to what the text implies, Table 2 does not mention values for Saturn.

13. Fig. 10, caption, and l. 382. What is PUMA-S with respect to PUMA, introduced earlier ?

14. It would be preferable to define the Burger number when it is first introduced (l. 241) rather than later on (Eq. 13).

15. L. 125. Say that $u^*$ and $v^*$ are perturbations with respect to zonal mean, and that the overbar denotes a longitudinal mean.

16. Table 1 does not seem to be referenced in the text. It could be on l. 363, after mention of the range of variation of $\Omega^*$.

17. L. 350, $\Delta\theta_E P \rightarrow \Delta\theta_{EP}$

18. Caption of Table 2. Expand *PDS (Planetary Data System)*

19. L. 64, … *quasi-geostrophic potential vorticity (**QGPV**), …*

20. L. 468, expand *LMD (Laboratoire de Météorologie Dynamique)*

21. L. 451, … *tilt with **altitude**.*

22. L. 141, … *yet **it** is observed …*

23. L. 114, … *stability criterion (i)* (parentheses, a similar correction is to be made in other places, please check)

---

## Referee Comment (RC1) · Anonymous Referee #1 · 19 Nov 2019

Fig 7. from Young (2014) is incorrect for FZ: 2.08 to be replaced by 1.69

Fig.8. Please define the Peclet number Pe (Nu-1 ?)

3.1 and Fig.9. Saturn and Jupiter do have super-rotation in the equator (and multiple jets, Note that Saturn "S" is missing in Fig.9) in contradiction with Fig.9. This part seems to suggest that only large Rossby numbers yield super-rotation: this is not what is observed. It might be the case that the authors mean "global (?)" super-rotation and arguing that one is within the PUMA realm but a comment on this would be very welcome.

Fig 12. The arrow thicknesses must be consistent with their numerical values. The convention of using negative values is also misleading. Why not use positive values

provided the arrows are in the right direction ?

12(a) NCEP data are used here: plz correct -0.48 to 0.48 since the arrow direction is in the opposite direction. Note that one should look at Boer and Lambert instead of Fig. 1 of Tabataba-Vakili et al. (pls correct text as well).

12(b) pls refer explicitly to Fig.12 (b) in 4.1.2

12(c): correct to positive values like in Lee and Richardson.

---

## Referee Comment (RC2) · Anonymous Referee #2 · 1 Jan 2020

This is an interesting and generally well-written review of barotropic and baroclinic instabilities, and their equilibration, in planetary atmospheres. I have mostly fairly minor comments and clarifications for the authors to consider.

1. I think it was Margules who introduced the idea of a potential energy that is available for conversion to kinetic energy. Granted it was Lorenz who (as the authors describe) came up with the eponymous energy cycle, but in a review article such as this it might be appropriate to reference Margules.

2. Regarding baroclinic equilibration and baroclinic adjustment. It is fairly apparent that Earth's ocean is in a supercritical state, not one of marginal criticality. The ocean may
be beyond the scope of this article, but that observation prompted a somewhat more general discussion of why supercriticality does occur that may be found in Jansen and Ferrari in 2012/2013. Going back a little further, precursors may be found in Salmon (GAFD 1980) and Vallis (QJ 1988) who both found supercritical regimes. The authors may wish to consider a brief discussion of all this.

3. Regarding Venus, it seems to me rather an 'ill-posed planet' as regards quasigeostrophy, or at least its troposphere is. Both N and f are close to zero (compared to values on Earth) so the deformation radius, NH/f (or equivalently the square root of the Burger number) is poorly defined. At cloud level N is finite but f is still small, so the conventional deformation radius is very large, probably larger than the planet itself. The discussion toward the end of section 4.2 needs a little more clarification, since as defined by equation (13) the Burger number does not vary with the reference frame, unless f is being considered as changing, which is perhaps what the authors mean. But in any case this needs clarifying and a bit more discussion.

4. It is not just realistically forced models that struggle to get the key features of Venus. Different models with Held-Suarez forcing can give very different results, as I think some recent model intercomparisons have shown (although I don't have a reference to hand). I suspect this is at least in part because of the ability of models to conserve angular momentum, and the fact that models are sensitive to the ratio of N/f, which might vary among models.

5. The authors say, early in in the conclusions, that 'the distinction between baroclinic and barotropic instabilities is to a large extent an artificial one.' Is that really the case? Granted the CSP criterion covers both cases, but as the authors themselves say the difference is associated with whether the basic state has a store of potential energy that is converted to eddy kinetic energy, and this doesn't seem like a minor difference. And the Lorenz cycles for barotropic and baroclinic instabilities are quite different. If the authors are trying to make a profound statement that, in spite of these differences, the instabilities are really of the same type then I think they need to justify this more. If Interactive comment
not, I'd suggest they moderate that statement.

6. At the beginning of the section on Jupiter the wording suggests that there are a number of gas giants in the Solar System, whereas in fact there are only two, if ice giants are regarded as separate. This is made clearer later on, but a slight rephrasing to clarify might help.

7. Is it really the case that (line 581) 'the release of latent heat energy from condensation of water vapour plays an important role ...in the weather layer'? Do we know that or is that really just a conjecture? After all, the moist and dry adiabatic lapse rates are almost the same on Jupiter because of the hydrogen atmosphere, suggesting that moisture may have a limited effect. Granted there are arguments the other way (lightning is seen, there is a strong virtual temperature effect, some numerical simulations), but the importance otherwise of moisture seems to me an open question. If I am mistaken and it is a settled issue then the authors need to point to some definitive evidence and give references.

---

## Author Comment (AC2) · 13 Jan 2020

Thank you for these comments and your detailed and careful reading of this manuscript. We will take these into account in producing a revised version of the paper, which we hope will then be acceptable. Our detailed responses follow.

*The line numbers below are those of the version I have had access to, once printed. For some reason, the numbers seem to be shifted by one or two units from the ones on the version I visualize on my screen.*

I have noticed this too! But I think most of your references are clear.

*1. Eq. (1). An integral sign (from $0$ to $H$ I presume) is missing for the integral with respect to $z$ on the left-hand side of the equation. And it might be useful to specify at*

[Figure]

*this stage that $y$ is the latitudinal coordinate and $z$ the vertical coordinate.*

Well spotted! Yes, the integral sign should be added. We will also specify the coordinates here as suggested.

*2. There seems to be an inconsistency as concerns the values of the thermal Rossby number $Ro_T$. The text (starting l. 388) says that the eddy meridional heat transport peaks at $Ro_T \sim 0.07$, while Fig. 11(a) shows a peak at $Ro_T \sim 0.3$ (see also Table 1, and ll. 724–730 and 748–751).*

This looks like a straightforward error – it will be corrected to $Ro_T \sim 0.3$ for the peak.

*3. Fig. 3(c). There are two curves on the figure. What is the difference between them ?*

This is just a single curve that is double valued, and comes from plotting $u$ vs d$Q$/d$y$ point wise across the domain represented in the radial profiles in 3(a) and (b). $u$ takes different values at different radii, despite have the same value of d$Q$/d$y$. This is presumably because the flow is not precisely at the inviscid marginally stable state.

*4. Ll. 444-445, ... leads to a value of $\mathcal{T}_R$ which is much larger than O(1) but not hugely so .... Well, the value given in Table 2 is 1.3 x $10^5$.*

OK point taken. We will omit the "but not hugely so" phrase.

*5. Ll. 710-711, ... to infer the existence of a unique reference frame on each planet ... A unique reference frame with which properties ?*

This is the unique state at which the gravest Doppler-shifted Rossby wave trains are just able to propagate at the same phase speed and hence couple together and interact to grow via over-reflection. We will clarify in the text.

*6. Ll. 461-462, ... the observed near suppression of baroclinic instability in Martian summers .... It would be better to give appropriate reference(s).*

OK. A reference will be inserted here.

*7. Fig. 6. What is the precise connection between the vertical coordinate (Stability parameter) and the thermal Rossby number ?*

OK - they are the same in this diagram (a consequence of using a figure from another source). We will clarify this in the caption.

*8. Fig. 8. Inset. It would be preferable to say explicitly that that Pe refers to heat transport by the axisymmetric flow, Pxs to transport by the eddies, and m to the number of longitudinal waves.*

This will be clarified.

*9. L. 100, ... for comparison, in Section 4, with the known properties ...*

Well spotted! Reference to Section 4 will be added.

*10. Eq. (15) Inconsistency of notation. $\theta$ or $\theta$ with overbar ?*

OK, overbar is intended here and will be added.

*11. Ll. 418-419, ... the values of Bu, $Ro_T$ and $\mathcal{T}_R$ [...], based on Eqs (12-14)*

Noted.

*12. Ll. 780-781. Contrary to what the text implies, Table 2 does not mention values for Saturn.*

Reference to Saturn will be deleted.

*13. Fig. 10, caption, and l. 382. What is PUMA-S with respect to PUMA, introduced earlier ?*

This was used in Wang et al. (2018) to mean the Held-Suarez simplified version of the PUMA model. But this distinction is unnecessary in the present context so the "-S" suffix will be omitted.

*14. It would be preferable to define the Burger number when it is first introduced (l. 241) rather than later on (Eq. 13).*

OK, the definition of Bu will be moved to this point.

*15. L. 125. Say that u\* and v\* are perturbations with respect to zonal mean, and that the overbar denotes a longitudinal mean.*

These definitions will be added.

*16. Table 1 does not seem to be referenced in the text. It could be on l. 363, after mention of the range of variation of $\Omega^*$.*

Noted - thanks for the suggestion.

*17. L. 350, $\Delta\theta_{EP} \rightarrow \Delta\theta_E P$*

Noted - to be corrected.

*18. Caption of Table 2. Expand PDS (Planetary Data System)*

Will do.

*19. L. 64, ... quasi-geostrophic potential vorticity (QGPV), ...*

We will add this.

*20. L. 468, expand LMD (Laboratoire de Météorologie Dynamique)*

Will do.

*21. L. 451, ... tilt with altitude.*

We propose to clarify this by "...pronounced latitudinal tilt with altitude".

*22. L. 141, ... yet it is observed ...*

OK – arguable but happy to change this.

*23. L. 114, ... stability criterion (i) (parentheses, a similar correction is to be made in other places, please check).*

Well spotted. Will check for other occurrences.
* * *

---

## Author Response (AR1)

**1 Introduction**

Further to the reviews of this submission, we have considered all of the comments made by both reviewers and Editor and taken them fully into account in revising our manuscript. We have accepted most of them and respond to each comment as follows. In addition, we include a version of the revised manuscript with significant changes highlighted in blue.

**2 Reviewer 1**

We are grateful to the referee for these comments and suggestions. We have implemented most of them in the text and figures, as indicated below, with changes to the text noted in blue in the new version of the manuscript.

(1) *Fig 7. from Young (2014) is incorrect for FZ: 2.08 to be replaced by 1.69*

Now done

(2) *Fig.8. Please define the Peclet number Pe (Nu-1 ?)*

Péclet number is indeed defined as Nu-1 (advective/conductive transport). This is now defined in the text.

(3) *3.1 and Fig.9. Saturn and Jupiter do have super-rotation in the equator (and multiple jets, Note that Saturn "S" is missing in Fig.9) in contradiction with Fig.9. This part seems to suggest that only large Rossby numbers yield super-rotation: this is not what is observed. It might be the case that the authors mean "global (?)" super-rotation and arguing that one is within the PUMA realm but a comment on this would be very welcome.*

Global super-rotation was intended in this discussion and a comment has been added to clarify this (with a reference to the review by Read & Lebonnois 2018). We also note that Jupiter and Saturn have locally super-rotating equatorial jets, although the global super-rotation is not known for either planet. The reference (S) to Saturn in Fig. 9 has been added.

(4) *Fig 12. The arrow thicknesses must be consistent with their numerical values. The convention of using negative values is also misleading. Why not use positive values provided the arrows are in the right direction ?*

Noted. We have now changed the convention to use positive numbers everywhere and have adjusted the thickness of arrows to be clearer.

(5) *12(a) NCEP data are used here: plz correct -0.48 to 0.48 since the arrow direction is in the opposite direction. Note that one should look at Boer and Lambert instead of Fig. 1 of Tabataba-Vakili et al. (pls correct text as well).*

Number changed to positive in Fig. 12(a) and references are now entirely to Boer & Lambert for these numbers.

(6) *12(b) pls refer explicitly to Fig.12 (b) in 4.1.2*

Now added.

(7) *12(c) correct to positive values like in Lee and Richardson.*

Done.

**3  Reviewer 2**

*This is an interesting and generally well-written review of barotropic and baroclinic instabilities, and their equilibration, in planetary atmospheres. I have mostly fairly minor comments and clarifications for the authors to consider.*

Thanks for this. We respond to these points below and have modified the text of the paper as indicated.

(1) *I think it was Margules who introduced the idea of a potential energy that is available for conversion to kinetic energy. Granted it was Lorenz who (as the authors describe) came up with the eponymous energy cycle, but in a review article such as this it might be appropriate to reference Margules.*

Noted. We have added a reference to Margules (1903) as suggested.

(2) *Regarding baroclinic equilibration and baroclinic adjustment. It is fairly apparent that Earth's ocean is in a supercritical state, not one of marginal criticality. The ocean may be beyond the scope of this article, but that observation prompted a somewhat more general discussion of why supercriticality does occur that may be found in Jansen and Ferrari in 2012/2013. Going back a little further, precursors may be found in Salmon (GAFD 1980) and Vallis (QJ 1988) who both found supercritical regimes. The authors may wish to consider a brief discussion of all this.*

This is a good point. Although the ocean is probably beyond the intended scope of the article, similar super-critical regimes may occur in gas giant atmospheres. We have added some extra discussion of this in Section 5, including mention of some of the scaling arguments of Jansen and Ferrari. A notable addition to this from the present work is to emphasise the non-negligible role of the zonally-symmetric meridional overturning in addition to eddy transports in maintaining a criticality close to 1 under some conditions. This is now clarified in our discussion.

(3) *Regarding Venus, it seems to me rather an 'ill-posed planet' as regards quasigeostrophy, or at least its troposphere is. Both N and f are close to zero (compared to values on Earth) so the deformation radius, NH/f (or equivalently the square root of the Burger number) is poorly defined. At cloud level N is finite but f is still small, so the conventional deformation radius is very large, probably larger than the planet itself. The discussion toward the end of section 4.2 needs a little more clarification, since as*

*defined by equation (13) the Burger number does not vary with the reference frame, unless f is being considered as changing, which is perhaps what the authors mean. But in any case this needs clarifying and a bit more discussion.*

Noted. We have clarified in our discussion in Section 4.2 that the modified Burger number in the cloud layers assumes a change in $f$ to a value representative of the mean angular velocity of the cloud layer itself - with a rotation period of around 4 Earth days. This is what has been assumed in other published work, such as that by Young et al. (1984) and Sugimoto et al. (2014).

(4) *It is not just realistically forced models that struggle to get the key features of Venus. Different models with Held-Suarez forcing can give very different results, as I think some recent model intercomparisons have shown (although I don't have a reference to hand). I suspect this is at least in part because of the ability of models to conserve angular momentum, and the fact that models are sensitive to the ratio of N/f, which might vary among models.*

Agreed. We have included some additional discussion of idealised models of the Venus and Titan atmospheres and now reference the recent intercomparisons of such models by Lebonnois and Lee & Richardson.

(5) *The authors say, early in in the conclusions, that 'the distinction between baroclinic and barotropic instabilities is to a large extent an artificial one.' Is that really the case? Granted the CSP criterion covers both cases, but as the authors themselves say the difference is associated with whether the basic state has a store of potential energy that is converted to eddy kinetic energy, and this doesn't seem like a minor difference. And the Lorenz cycles for barotropic and baroclinic instabilities are quite different. If the authors are trying to make a profound statement that, in spite of these differences, the instabilities are really of the same type then I think they need to justify this more. If not, I'd suggest they moderate that statement.*

Point taken. The similarities are from a vorticity point of view only, but the distinction remains energetically as noted here. We have clarified this in Section 5.

(6) *At the beginning of the section on Jupiter the wording suggests that there are a number of gas giants in the Solar System, whereas in fact there are only two, if ice giants are regarded as separate. This is made clearer later on, but a slight rephrasing to clarify might help.*

OK though the dynamical distinction between gas and ice giants is perhaps rather a fine and subtle one. Section 4.3 does make a distinction between gas and ice giants in a few places, but we have added emphasis to the distinction in our discussion of Jupiter.

(7) *Is it really the case that (line 581) 'the release of latent heat energy from condensation of water vapour plays an important role ...in the weather layer'? Do we know that or is that really just a conjecture? After all, the moist and dry adiabatic lapse rates are almost the same on Jupiter because of the hydrogen atmosphere, suggesting that moisture may have a limited effect. Granted there are arguments the other way (lightning is seen, there is a strong virtual temperature effect, some numerical simulations), but the importance otherwise of moisture seems to me an open question. If I am mistaken and it is a settled issue then the authors need to point to some definitive evidence and give references.*

Precisely what the role of latent heat release in moist convection is in the weather layers of Jupiter and Saturn is still an open one. But evidence for the occurrence of moist convection I think is fairly unequivocal (notably through observations of lightning and convective water clouds). Also, estimates of upward energy flux by moist convection (e.g. by Gierasch et al. (2000)) suggest that an O(1) fraction of the interior heat on Jupiter ($> 50\%$) may be carried by moist convective processes. This led Ingersoll et al. (2000) to suggest that the dominant eddies driving the zonal jets might also be from moist convection, although the recent work of Young and Read (2017) suggests that baroclinic instabilities (on scales similar to the Rossby deformation radius) may be most important for this. We have added some discussion in Section 4.3 to clarify this.

**4 Editor**

Thank you for these comments and your detailed and careful reading of this manuscript. We will take these into account in producing a revised version of the paper, which we hope will then be acceptable. Our detailed responses follow.

- *The line numbers below are those of the version I have had access to, once printed. For some reason, the numbers seem to be shifted by one or two units from the ones on the version I visualize on my screen.*

  I have noticed this too! But I think most of your references are clear.

(1) *1. Eq. (1). An integral sign (from 0 to H I presume) is missing for the integral with respect to z on the left-hand side of the equation. And it might be useful to specify at this stage that y is the latitudinal coordinate and z the vertical coordinate.*

  Well spotted! Yes, the integral sign has been added. We also specify the coordinates here as suggested.

(2) *There seems to be an inconsistency as concerns the values of the thermal Rossby number $Ro_T$. The text (starting l. 388) says that the eddy meridional heat transport peaks at $Ro_T \sim 0.07$, while Fig. 11(a) shows a peak at $Ro_T \sim 0.3$ (see also Table 1, and ll. 724–730 and 748–751).*

A straightforward error – now corrected to $Ro_T \sim 0.3$ for the peak.

(3) *Fig. 3(c). There are two curves on the figure. What is the difference between them ?*

This is just a single curve that is double valued, and comes from plotting $u$ vs $dQ/dy$ point wise across the domain represented in the radial profiles in 3(a) and (b). $u$ takes different values at different radii, despite have the same value of $dQ/dy$. This is presumably because the flow is not precisely at the inviscid marginally stable state, though we don't think it is necessary to labour this point in the text.

(4) *Ll. 444-445, ... leads to a value of $\mathcal{T}_R$ which is much larger than O(1) but not hugely so .... Well, the value given in Table 2 is 1.3 x $10^5$.*

OK point taken. We have omitted the "but not hugely so" phrase.

(5) *Ll. 710-711, ... to infer the existence of a unique reference frame on each planet ... A unique reference frame with which properties ?*

This is the unique state at which the gravest Doppler-shifted Rossby wave trains are just able to propagate at the same phase speed and hence couple together and interact to grow via over-reflection. We have added text in Section 5.2 to clarify this.

(6) *Ll. 461-462, ... the observed near suppression of baroclinic instability in Martian summers .... It would be better to give appropriate reference(s).*

OK. A reference to the paper by Lewis et al. (2016) has been inserted here.

(7) *Fig. 6. What is the precise connection between the vertical coordinate (Stability parameter) and the thermal Rossby number ?*

OK - they are the same in this diagram (a consequence of using a figure from another source). We have clarified this in the caption.

(8) *Fig. 8. Inset. It would be preferable to say explicitly that that Pe refers to heat transport by the axisymmetric flow, Pxs to transport by the eddies, and m to the number of longitudinal waves.*

This has been clarified in the caption.

(9) *L. 100, ... for comparison, in Section 4, with the known properties ...*

Well spotted! Reference to Section 4 has been added.

(10) *Eq. (15) Inconsistency of notation. $\theta$ or $\theta$ with overbar ?*

OK, overbar is intended here and has been added.

(11) *Ll. 418-419, ... the values of Bu, $Ro_T$ and $\mathcal{T}_R$ [...], based on Eqs (12-14)*

Noted and corrected.

(12)  *Ll. 780-781. Contrary to what the text implies, Table 2 does not mention values for Saturn.*

Reference to Saturn has been deleted.

(13)  *Fig. 10, caption, and l. 382. What is PUMA-S with respect to PUMA, introduced earlier ?*

This was used in Wang et al. (2018) to mean the Held-Suarez simplified version of the PUMA model. But this distinction is unnecessary in the present context so the "-S" suffix has been omitted throughout the paper.

(14)  *It would be preferable to define the Burger number when it is first introduced (l. 241) rather than later on (Eq. 13).*

OK, the definition of Bu has been added here (as well as the modified definition in Eq (14) in the revised version (line 350).

(15)  *L. 125. Say that u\* and v\* are perturbations with respect to zonal mean, and that the overbar denotes a longitudinal mean.*

These definitions have been added.

(16)  *Table 1 does not seem to be referenced in the text. It could be on l. 363, after mention of the range of variation of $\Omega^*$.*

Noted - thanks for the suggestion. Now added.

(17)  *L. 350, $\Delta\theta_{EP} \rightarrow \Delta\theta_E P$*

Noted - now corrected.

(18)  *Caption of Table 2. Expand PDS (Planetary Data System)*

Now added.

(19)  *L. 64, ... quasi-geostrophic potential vorticity (QGPV), ...*

Now added.

(20)  *L. 468, expand LMD (Laboratoire de Météorologie Dynamique)*

Now added.

(21)  *L. 451, ... tilt with altitude.*

We have clarified this by "...pronounced latitudinal tilt with altitude".

(22)  *L. 141, ... yet it is observed ...*

Now changed.

(23)  *L. 114, ... stability criterion (i) (parentheses, a similar correction is to be made in other places, please check).*

Well spotted. Now corrected in this and a few other places.

[revised manuscript text omitted]